# Error Propagation in Dynamic Programming: From Stochastic Control to Option Pricing

## Abstract

This paper investigates theoretical and methodological foundations for stochastic optimal control (SOC) in discrete time. We start formulating the control problem in a general dynamic programming framework, introducing the mathematical structure needed for a detailed convergence analysis. The associate value function is estimated through a sequence of approximations combining nonparametric regression methods and Monte Carlo subsampling. The regression step is performed within reproducing kernel Hilbert spaces (RKHSs), exploiting the classical KRR algorithm, while Monte Carlo sampling methods are introduced to estimate the continuation value. To assess the accuracy of our value function estimator, we propose a natural error decomposition and rigorously control the resulting error terms at each time step. We then analyze how this error propagates backward in time-from maturity to the initial stage-a relatively underexplored aspect of the SOC literature. Finally, we illustrate how our analysis naturally applies to a key financial application: the pricing of American options.

## 1 Introduction and Related Work

Stochastic optimal control (SOC) provides a principled framework for sequential decision-making under uncertainty. It plays a foundational role in a wide range of scientific and engineering domains, including economics and finance (Fleming & Stein, 2004; Pham, 2009; Åström, 2012), robotics (Gorodetsky et al., 2018; Theodorou et al., 2011), molecular dynamics (Hartmann & Schütte, 2012; Hartmann et al., 2013; Zhang et al., 2014; Holdijk et al., 2023), and stochastic filtering and data assimilation (Mitter, 2002; Reich, 2019). More recently, SOC has inspired advances in machine learning, particularly in tasks such as sampling from unnormalized distributions (Zhang & Chen, 2021; Berner et al., 2022; Richter & Berner, 2023; Vargas et al., 2023), nonconvex optimization (Chaudhari et al., 2018), optimal transport (Villani et al., 2008), and the numerical solution of backward stochastic differential equations (BSDEs) (Carmona, 2016).

Despite the extensive literature on continuous-time SOC (Bertsekas, 2012), its discrete-time counterpart remains highly relevant in practice, as it naturally arises in computational and data-driven settings where decisions are made at fixed time intervals (Bertsekas & Shreve, 1996; Hernández-Lerma & Lasserre, 2012; Puterman, 2014). At the same time, discrete-time SOC poses distinct challenges, largely because many of the analytical tools available in continuous time are no longer applicable. Nevertheless, it still offers opportunities for the development of scalable numerical methods, particularly through dynamic programming and function approximation. Discrete-time SOC is central to modern applications in operations research, financial engineering or reinforcement learning (RL) (Sutton et al., 1998). At its core lies a dynamic programming (DP) recursion, where the value function is computed backward in time via the Bellman operator (Bellman, 1966). In high-dimensional settings, solving this recursion exactly is often infeasible, inspiring a large body of research focused on developing scalable and efficient approximations. These approaches typically estimate value functions from data using simulation or function approximation. In recent years, deep learning has greatly expanded the scalability of these methods, enabling their application to high-dimensional control problems (Han et al., 2016; Domingo i Enrich et al., 2024).

Despite this empirical progress, the theoretical understanding of learning-based SOC remains limited. A key challenge lies in quantifying how local errors deriving from function approximation, sampling noise, or optimization inaccuracies, propagate through the Bellman recursion over time.

Studying this requires a rigorous and principled mathematical framework in which to analyze error accumulation in high-dimensional value function approximations. In this work, we propose such a framework based on reproducing kernel Hilbert spaces (RKHS), which enables us to derive explicit error bounds and control error propagation in approximate dynamic programming.

A classical application of discrete-time SOC is the pricing of American-style options, also known as Bermudan options when exercise opportunities are discrete. This problem can be formulated as a finite-horizon optimal stopping problem under stochastic dynamics. While such problems can, in principle, be solved exactly (Peskir & Shiryaev, 2006; Lamberton & Lapeyre, 2011), well-established numerical methods, such as tree-based approaches or PDE solvers, struggle with the curse of dimensionality as complexity increases (Broadie & Glasserman, 1997; Bally et al., 2003; Jain & Oosterlee, 2012). Monte Carlo–based methods have become widespread in high-dimensional applications. Notable examples include regression-based techniques (Tsitsiklis & Van Roy, 1999; Longstaff & Schwartz, 2001), dual and hybrid primal-dual formulations (Rogers, 2002; Haugh & Kogan, 2004; Andersen & Broadie, 2004; Belomestny et al., 2013; Lelong, 2018), and Malliavin calculus methods for estimating conditional expectations (Lions & Regnier, 2001; Bouchard & Touzi, 2004; Bally et al., 2005; Abbas-Turki & Lapeyre, 2012). More recently, machine learning (Williams & Rasmussen, 2006) and deep learning approaches (Kohler et al., 2010; Nielsen, 2015; Becker et al., 2019; Goudenege et al., 2020) have shown strong empirical performance in this domain. However, these methods often lack rigorous theoretical guarantees on accuracy and generalization.

Our work aims to bridge this gap by developing kernel-based algorithms for discrete-time SOC that come with provable convergence guarantees and theoretical error bounds, while keeping an eye on computational efficiency and scalability for big data applications.

**Contribution** In summary, our contributions are as follows. First, we propose a general RKHS-based formulation of approximate dynamic programming through backward induction. Second, we provide a rigorous and transparent decomposition of the total approximation error into three distinct components: regression error, Monte-Carlo sampling error, and propagation error. Third, we derive explicit convergence rates under model misspecification by leveraging source conditions. Finally, we show how this framework can be applied to various problems, especially in finance. We demonstrate the practical effectiveness of our algorithm through the well-known problem of American option pricing and preliminarily test its performance against some of the standard benchmark methods in the field.

**Organization** The paper is organized as follows. In Section 2 we introduce the problem and setting, with key definitions and notations used throughout the paper, and formalizing the problem in a precise mathematical framework. In Section 3 we introduce the Monte Carlo approximation and the regression step in the RHKS environment. In Section 4 we study the error back-propagation, upper bounding the various approximation terms and finally showing the final error guarantees in Theorem 1. In Section 5 we finally present some numerical results.

## 2 SETTING AND STOCHASTIC CONTROL MODEL

Consider a discrete time horizon $t = \{0, 1, \ldots, T\}$. We define a stochastic process $Z \coloneqq (Z_t)_{t=0}^T$ on a filtered probability space $(\Omega, \mathcal{F}, (\mathcal{F}_t^Z)_{t=0}^T, \mathbb{P})$, where $(\mathcal{F}_t^Z)_{t=0}^T$ is the natural filtration generated by $Z$. The random variables $Z_t$ are mutually independent (but not necessarily identically distributed) and take values in measurable spaces $\mathcal{Z}_t$. We denote by $\mathbb{P}(dz) = \prod_{t=0}^T \mathbb{P}_t(dz_t)$ the distribution of $Z$ on the path space $\mathcal{Z} = \mathcal{Z}_0 \times \cdots \times \mathcal{Z}_T$, which we identify with $\Omega$ without loss of generality. Any square-integrable adapted process, such as an asset price process, can be written in the form $X_t = X_t(Z_0, \ldots, Z_t)$, for some function $X_t \in L^2_{\mathbb{P}_0 \times \cdots \times \mathbb{P}_t}$.

Controlled Markov process $X_0^u, \ldots, X_T^u$ taking values in state spaces $\mathcal{X}_0, \ldots, \mathcal{X}_T$ is defined by

$$\begin{cases} X_0^u = p_0(Z_0), \\ X_{t+1}^u = \pi_t\big(X_t^u, u_t(X_t^u), Z_{t+1}\big), \quad t \in \{0, \ldots, T-1\}, \end{cases} \tag{1}$$

where $p_0 : \mathcal{Z}_0 \to \mathcal{X}_0$ is the initial state distribution, $\pi_t : \mathcal{X}_t \times \mathcal{U}_t \times \mathcal{Z}_{t+1} \to \mathcal{X}_{t+1}$ is a Markov transition function encoding how the system transitions from one state to the next, and $\boldsymbol{u} = (u_t)_{t=0}^{T-1} \in \boldsymbol{\mathcal{U}}$ is a stochastic control law, where each $u_t : \mathcal{X}_t \to \mathcal{U}_t$ is $\mathcal{F}_t$-measurable.

**Remark 1.** *Note that this setup remains very general despite the Markovian assumption. For reasons related to dimensionality, it is typically assumed that $X_t^u$ summarizes the full history $Z_{0:t}, u_{0:t}$ in a compressed form, so that the control at time $t$ depends only on $X_t^u$, i.e., $u_t = u_t(X_t^u)$. This does not entail a loss of generality, as many important problems are naturally Markovian. Moreover, any optimal stopping problem can be cast in Markovian form by including all relevant past information in the current state, at the cost of increasing dimensionality.*

A control law $\boldsymbol{u}$ is said to be *admissible* if the maps $(x, z) \mapsto \pi_t(x, u_t(x), z)$ satisfy suitable regularity conditions. In particular, we assume that the operator

$$P_t^u f(x) := \mathbb{E}\left[ f(X_{t+1}^u) \mid X_t^u = x \right] = \mathbb{E}\left[ f\big(\pi_t(x, u, Z_{t+1})\big) \right] = \int_{\mathcal{Z}_{t+1}} f\big(\pi_t(x, u, z)\big) \mathbb{P}_{t+1}(\mathrm{d}z), \tag{2}$$

defines a Markov transition kernel from $\mathcal{X}_t$ to $\mathcal{X}_{t+1}$, for all $u \in \mathcal{U}_t$. With a slight abuse of notation, we will sometimes use the alternative, also common definition in kernel form

$$P_t^u(x, A) := \mathbb{P}\left[ X_{t+1}^u \in A \mid X_t^u = x \right] = \int_{\mathcal{Z}_{t+1}} \mathbb{1}_A(\pi_t(x, u, z)) \mathbb{P}_{t+1}(\mathrm{d}z) \tag{3}$$

with $A \in \mathcal{B}(\mathcal{X}_{t+1})$, i.e. the Borel $\sigma$-algebra on the space $\mathcal{X}_{t+1}$. In the following, it will be clear which one of the two representations we are using. The connection between the two is simply

$$P_t^u f(x) = \int_{\mathcal{X}_{t+1}} f(x') P_t^u(x, \mathrm{d}x'). \tag{4}$$

The objective of stochastic optimal control is to maximize a *gain* function over all admissible control laws. In the discrete-time setting, this is given by the sum of the partial rewards $F_t : \mathcal{X}_t \times \mathcal{U}_t \to \mathbb{R}$ for $t = 0, \ldots, T-1$, and the terminal reward $\Phi = F_T : \mathcal{X}_T \to \mathbb{R}$. Then, we define the optimal value function $V_t : \mathcal{X}_t \to \mathbb{R}$ at time $t$ as

$$V_t := \sup_{\boldsymbol{u} \in \mathcal{U}} \mathbb{E}\left[ \sum_{s=t}^{T-1} F_s\left(X_s^u, u_s(X_s^u)\right) + \Phi\left(X_T^u\right) \mid X_t^u \right]. \tag{5}$$

We now introduce the Bellman operator at time $t$ as

$$\mathcal{T}_t f(x) := \operatorname*{ess\,sup}_{u \in \mathcal{U}_t} F_t(x, u) + P_t^u f(x). \tag{6}$$

Bellman's principle (Bellman, 1966) implies that the optimal value function solves the dynamic programming equation (Bertsekas & Shreve, 1996; Kallsen, 2016)

$$\begin{cases} V_T(x) = \Phi(x), \\ V_t(x) = \mathcal{T}_t V_{t+1}(x), \qquad t \in \{0, \ldots, T-1\}. \end{cases} \tag{7}$$

We now want to represent Eq. 7 as a functional dynamic programming equation in some appropriate $L^2$ spaces. To this end, we fix an auxiliary admissible control law $\bar{u}$, often called the behavior policy in the RL literature (Sutton et al., 1998), and let $\mu_t$ denote the distribution of $X_t^{\bar{u}}$ on $\mathcal{X}_t$. We introduce the following assumption to ensure that, if $\Phi \in L^2_{\mu_T}$, then the optimal value function satisfying the dynamic system 7 belongs to $L^2_{\mu_t}$ for all $t \in \{0, \ldots, T\}$.

**Assumption 1** (Square integrability). *There exist constants $c_F > 0$ and $c_P > 0$ such that, for all $t \in \{0, \ldots, T\}$:*

$$\left\| \operatorname*{ess\,sup}_{u \in \mathcal{U}_t} |F_t(\cdot, u)| \right\|_{L^2_{\mu_t}} \leqslant c_F, \qquad \left\| \operatorname*{ess\,sup}_{u \in \mathcal{U}_t} |P_t^u g| \right\|_{L^2_{\mu_t}} \leqslant c_P^{1/2} \|g\|_{L^2_{\mu_{t+1}}}, \tag{8}$$

*for all $g \in L^2_{\mu_{t+1}}$. We further assume $\Phi \in L^2_{\mu_T}$.*

Under these conditions, $\mathcal{T}_t : L^2_{\mu_{t+1}} \to L^2_{\mu_t}$, see Lemma 2 in Appendix A for the complete proof. Then $V_t \in L^2_{\mu_t}$ for all $t = \{0, \ldots, T\}$, and the dynamic programming problem 7 holds in each corresponding $L^2_{\mu_t}$ space. Further details on this assumption are discussed in Appendix D.

The main goal in the following will be to find a good estimate of the optimal value function at the initial time $t = 0$, i.e. $V_0$, by leveraging the recursive formulation in 7.

**Example 1** (American Options). *An American option is a financial contract that gives the holder the right, but not the obligation, to buy or sell an underlying asset at a specified strike price at any time up to the expiration date.*

*Let $X$ be an exogenous Markov process, i.e., it is not influenced by any control variable or decision. Let $Q_t = Q_t(x, dx')$ denote its Markov transition kernel from $\mathcal{X}_t$ to $\mathcal{X}_{t+1}$, which specifies the conditional distribution of the next state $X_{t+1}$ given the current state $X_t = x$. Suppose the underlying asset has a price at time $t$ given by a function $S_t(X_t)$. An American (call) option with strike $K$ pays*

$$C_t(X_t) = (S_t(X_t) - K)^+ \tag{9}$$

*if exercised at time $t$. In practice, the dimension of the state space can be very high. For instance, $S_t(X_t)$ could represent the maximum price in a basket of assets at time $t$, as in a so-called American max-call option.*

*The holder of the American option aims to maximize the expected payoff $\mathbb{E}[C_\tau(X_\tau)]$ over all exercise strategies, i.e., over all stopping times $\tau$. We now cast this problem as a stochastic optimal control problem 7. To this end, we introduce a cemetery state $\Delta_\dagger \notin \mathcal{X}_t$ and define the augmented state space $\mathcal{X}_t^{\Delta_\dagger} := \mathcal{X}_t \cup \{\Delta_\dagger\}$. Any measurable function $f$ on $\mathcal{X}_t$ is extended to $\mathcal{X}_t^{\Delta_\dagger}$ by setting $f(\Delta_\dagger) := 0$. This is a standard technique in the theory of Markov processes, see Revuz & Yor (2013).*

*Define now the control space as $\mathcal{U}_t := \{0, 1\}$, where $u = 0$ represents exercising the option and $u = 1$ holding it. The controlled Markov transition kernel $P_t^u$ is given by:*

$$P_t^u(x, A) = \begin{cases} Q_t(x, A \cap \mathcal{X}_{t+1}) & \text{if } u = 1, \ x \in \mathcal{X}_t, \\ \delta_{\Delta_\dagger}(A) & \text{otherwise,} \end{cases}$$

*for $A \in \mathcal{B}(\mathcal{X}_{t+1}^\Delta)$, meaning that the controlled process $X_t^u$ follows the exogenous dynamics until the option is exercised, after which it is absorbed in the state $\Delta_\dagger$. Define also:*

$$F_t(\cdot, 1) = 0, \qquad F_t(\cdot, 0) = C_t, \qquad \Phi = C_T. \tag{10}$$

*An admissible control law $\mathbf{u}$ then consists of measurable functions $u_t : \mathcal{X}_t^{\Delta_\dagger} \to \{0, 1\}$, with the convention $u_t(\Delta_\dagger) = 0$. The associated exercise strategy is defined by:*

$$\tau := \inf\{t \mid u_t = 0\} \wedge T, \tag{11}$$

*i.e., the first time $t \in \{0, \dots, T-1\}$ such that $u_t = 0$, or $T$ if no such time exists (with $\inf \emptyset = \infty$). The dynamic programming problem 7 then becomes:*

$$\begin{cases} V_T(x) = C_T(x), \\ V_t(x) = \max\left\{C_t(x), e^{-r} Q_t V_{t+1}(x)\right\}, \end{cases} \tag{12}$$

*which selects the maximum between the immediate exercise value and the (discounted) continuation value. Here, $r$ denotes the risk-free interest rate.*

## 3   SAMPLE-BASED VALUE FUNCTION APPROXIMATION

Stochastic dynamic control problem 7 is not directly solvable in practice, primarily because we do not have access to the true expectation in $P_t^u$. A standard way to address this issue is to approximate the expectation via Monte Carlo simulation. Let $\{z_i^{(t+1)}\}_{i=1}^{M_t} \sim \mathbb{P}_{t+1}^{M_t}$ be i.i.d. samples from the distribution of the stochastic driver $Z_{t+1}$. We then define

$$\widetilde{P}_t^u f(x) := \frac{1}{M_t} \sum_{i=1}^{M_t} f\left(\pi_t(x, u, z_i^{(t+1)})\right), \qquad \widetilde{\mathcal{T}}_t f(x) := \operatorname*{ess\,sup}_{u \in \mathcal{U}_t} \left\{F_t(x, u) + \widetilde{P}_t^u f(x)\right\}, \tag{13}$$

the empirical approximation of $P_t^u$ and the associated empirical Bellman operator, respectively. By the Law of Large Numbers and the Continuous Mapping Theorem, we obtain:

$$\widetilde{P}_t^u f(x) \xrightarrow{a.s.} P_t^u f(x), \qquad \widetilde{\mathcal{T}}_t f(x) \xrightarrow{a.s.} \mathcal{T}_t f(x), \qquad \text{for } M_t \to \infty. \tag{14}$$

However, a naive application of this approximation—by recursively replacing $P_t^u$ with $\widetilde{P}_t^u$ in the dynamic programming equation—fails in practice, resulting in a nested Monte Carlo procedure whose

computational cost grows exponentially with $T$, making it infeasible for large time horizons.

To mitigate this, we adopt a more efficient approach: we proceed backward in time and use regression to construct a sequence of function approximators for each $V_t$. At each stage, we generate samples and solve a supervised learning problem, leveraging the approximation of $V_{t+1}$ obtained in the previous step (with the terminal condition $V_T = \Phi$ known a priori). Specifically, assume we have already computed an approximation of $V_{t+1}$, denoted by $\widehat{W}_{t+1}^{\lambda_{t+1}}$ (this $\widehat{\cdot}$ notation will be explained below in Eq. 18). We then generate training data $\{(x_i, y_i)\}_{i=1}^{n_t}$, where $x_i \sim \mu_t$ and

$$y_i = \widetilde{\mathcal{T}_t} \widehat{W}_{t+1}^{\lambda_{t+1}}(x_i). \tag{15}$$

We now solve the corresponding regression problem using a suitable supervised learning method. A classical choice is *regularized empirical risk minimization* (ERM) with Tikhonov regularization. Combined with kernel methods, and with the natural choice of the square loss as the loss function, this yields the well-known *Kernel Ridge Regression* (KRR).

**Assumption 2** (Reproducing Kernel Hilbert Space). *Let $\mathcal{H}_k$ be a separable reproducing kernel Hilbert space (RKHS) of real-valued functions on $\mathcal{X}$, with inner product $\langle \cdot, \cdot \rangle_{\mathcal{H}_k}$ and associated norm $\| \cdot \|_{\mathcal{H}_k}$. Let $k : \mathcal{X} \times \mathcal{X} \to \mathbb{R}$ be the reproducing kernel of $\mathcal{H}_k$ and assume it is bounded, i.e., there exists $\kappa > 0$ such that $\sup_{x \in \mathcal{X}} k(x, x) \leqslant \kappa^2$.*

**Remark 2.** *Although we use standard well-spread Monte Carlo sampling in this step, this is not the only viable choice. Any quadrature rule (e.g., monomial rules) can be used in place of Eq. 13 to approximate the operator $P_t^u$. This flexibility can be especially valuable in high-dimensional settings or when Monte Carlo sampling error is non-negligible, as some quadrature methods may achieve much higher precision using fewer points.*

**KRR estimator.** For a regularization parameter $\lambda_t > 0$, the KRR estimator at time $t$ is defined as

$$\widehat{W}_t^{\lambda_t} := \arg\min_{f \in \mathcal{H}_k} \frac{1}{n_t} \sum_{i=1}^{n_t} (y_i - f(x_i))^2 + \lambda_t \|f\|_{\mathcal{H}_k}^2 \tag{16}$$

Note that at maturity, the value function $V_T$ is known and equals $\Phi$, so no approximation is needed at the final step. Also note that, given Eq. 15, $\widetilde{\mathcal{T}_t} \widehat{W}_{t+1}^{\lambda_{t+1}}$ is the regression target function, i.e.,

$$W_t^* := \widetilde{\mathcal{T}_t} \widehat{W}_{t+1}^{\lambda_{t+1}} = \arg\min_{f \in L_{\mu_t}^2} \mathbb{E}\left[ (Y - f(X))^2 \right] = \arg\min_{f \in L_{\mu_t}^2} \mathbb{E}\left[ \left( \widetilde{\mathcal{T}_t} \widehat{W}_{t+1}^{\lambda_{t+1}}(X) - f(X) \right)^2 \right] \tag{17}$$

since $\widetilde{\mathcal{T}_t} \widehat{W}_{t+1}^{\lambda_{t+1}} \in L_{\mu_t}^2$ under Assumption 1. In general, $W_t^* \notin \mathcal{H}_k$, i.e. the model is misspecified. We will mention this further in the next section when introducing the well-known *source condition*.

Before turning to the statistical analysis, we introduce a refinement of our estimator, which also justifies the notation $\widehat{\cdot}$ used above. This step will be important to control approximation errors in the next section. We recall the following definitions, see Chapter 6 in Steinwart & Christmann (2008a). Given a threshold parameter $B > 0$, we define the *clipped* version of $a \in \mathbb{R}$ as:

$$\widehat{a} := \min\{\max\{a, -B\}, B\}. \tag{18}$$

We say that a loss function $\ell$ is *clippable* at level $B > 0$ if for all $y \in \mathcal{Y}$ and $a \in \mathbb{R}$, $\ell(y, \widehat{a}) \leqslant \ell(y, a)$. It is easy to verify that many loss functions are clippable. In particular, the square loss (which we use) can be clipped at $B$ when the output $y \in [-B, B]$. Note that if $Y$ is generated as in Eq. 15, a sufficient condition for boundedness is $\sup_{x \in \mathcal{X}_t, u \in \mathcal{U}_t} |F_t(x, u)| < B$. In practice, $F_t(x, u)$ is often unbounded (e.g., option payoffs), but boundedness can be enforced without loss of rigor by restricting the dynamics to a compact subset of the state space. In financial applications, for instance, $\mu_t$ is typically induced by a discretized geometric Brownian motion, hence log-normal with exponentially decaying tails. Consequently, large deviations of $X_t$ are extremely rare, and truncation introduces only negligible error while allowing the use of the clipped estimator $\widehat{W}_t^{\lambda_t}$, as required in Steinwart & Christmann (2008b).

The resulting method–that for simplicity we will indicate as KRR-DP (Kernel Ridge Regression-Dynamic Programming) in the following–is summarized in Algorithm 1.

**Example 2** (American Options (cont.)). *We now return to the American options application introduced in Example 1, and continue adapting our model to this setting. Here, the state vector*

$X_t = (X_t^1, \ldots, X_t^d)^\top \in \mathbb{R}_+^d$ *represents the prices of $d$ underlying assets at time $t$. A common model for their evolution is geometric Brownian motion (GBM), whose dynamics are given by*

$$\mathrm{d}X_t^i = r X_t^i \, \mathrm{d}t + \sigma_i X_t^i \, (\rho^{1/2} \, \mathrm{d}B_t)^i, \tag{19}$$

*for $i = 1, \ldots, d$, with $r \in \mathbb{R}$ the risk-free rate, $\sigma_i > 0$ the volatility of asset $i$, $\rho \in \mathbb{R}^{d \times d}$ the correlation matrix and $B_t = (B_t^1, \ldots, B_t^d)^\top$ a d-dimensional Brownian motion with independent components. We consider discrete times $t = 0, \ldots, T$ and approximate the dynamics with*

$$X_{t+1}^i = X_t^i \cdot \exp\left( \left( r - \tfrac{1}{2}\sigma_i^2 \right) + \sigma_i \, (\rho^{1/2} z)_i \right), \tag{20}$$

*where $z = (z_1, \ldots, z_d)^\top \sim \mathcal{N}(0, I_d)$ is a vector of independent standard Gaussian variables. As an example, we define a max-call option with strike price $K > 0$, for which $S_t(X_t) = \max\{X_t^1, \ldots, X_t^d\}$, and the payoff at time $t$ is given by*

$$C_t(X_t) = (S_t(X_t) - K)^+ = \left( \max_{1 \leqslant i \leqslant d} X_t^i - K \right)^+. \tag{21}$$

*The transition function $\pi_t : \mathbb{R}_+^d \times \mathcal{U}_t \times \mathbb{R}^d \to \mathbb{R}_+^d$ is defined as*

$$\pi_t(x, u, z) := \begin{cases} \Delta_\dagger & \text{if } u = 0, \\ x \odot \exp\left( \left( r - \tfrac{1}{2}\sigma^2 \right) + \sigma \odot (\rho^{1/2} z) \right) & \text{otherwise,} \end{cases}$$

*with $\sigma = (\sigma_1, \ldots, \sigma_d)^\top$, $\odot$ the elementwise multiplication.*

---

**Algorithm 1:** KRR-DP for American Option Pricing (backward induction with MC + KRR)

---

**Inputs:** $T$; $r$; $\{C_t, \mu_t\, \pi_t, n_t, M_t\}_{t=0}^{T-1}$; KRR hyperparameters $\{\Theta_t\}_{t=0}^{T-1}$ (kernel, $\lambda_t$, etc.).

**Output:** Estimator $\widehat{W}_0^{\lambda_0} : \mathbb{R}^d \to \mathbb{R}$ of the value of the option $V_0$.

```
// MC estimate of discounted continuation under ``hold'' (u = 1)
```
**1 Function** ContinuationValue($x$, $M_t$, $\pi_t$, $\widehat{W}_{t+1}^{\lambda_{t+1}}$)**:**

**2** $\quad$ Sample $z^{(1)}, \ldots, z^{(M_t)} \overset{\text{i.i.d.}}{\sim} \mathbb{P}_{t+1}$; $\qquad\qquad$ `// e.g., z ~ N(0,I)`

**3** $\quad$ **for** $j = 1, \ldots, M_t$ **do**

**4** $\quad\quad$ $\widetilde{x}_j \leftarrow \pi_t(x, u{=}1, z^{(j)})$;

**5** $\quad$ **end**

**6** $\quad$ **return** $e^{-r\Delta t} \frac{1}{M_t} \sum_{j=1}^{M_t} \widehat{W}_{t+1}^{\lambda_{t+1}}(\widetilde{x}_j)$;

```
// Generate supervised data (X̂_t, ŷ_t) at stage t
```
**7 Function** DataGeneration($n_t$, $M_t$, $\mu_t$, $\pi_t$, $C_t$, $\widehat{W}_{t+1}^{\lambda_{t+1}}$)**:**

**8** $\quad$ Sample $\widehat{X}_t = [x_1, \ldots, x_{n_t}]^\top$, with $x_i \overset{\text{i.i.d.}}{\sim} \mu_t$;

**9** $\quad$ **parallel for** $i = 1, \ldots, n_t$ **do**

**10** $\quad\quad$ $q_i \leftarrow$ ContinuationValue($x_i$, $M_t$, $\pi_t$, $\widehat{W}_{t+1}^{\lambda_{t+1}}$); $\quad$ `// MC continuation`

**11** $\quad\quad$ $y_i \leftarrow \max\big(C_t(x_i), q_i\big)$; $\qquad$ `// Bellman: exercise vs. continue`

**12** $\quad$ **end**

**13** $\quad$ **return** $(\widehat{X}_t, \widehat{y}_t = [y_1, \ldots, y_{n_t}]^\top)$

```
// Main backward pass
```
**14 Function** OptionPricing($\{(n_t, M_t, \mu_t, \pi_t, C_t, \Theta_t)\}_{t=0}^T$)**:**

**15** $\quad$ $\widehat{W}_T^{\lambda_T} \leftarrow C_T \equiv \Phi$; $\qquad\qquad$ `// terminal value is known`

**16** $\quad$ **for** $t = T{-}1, \ldots, 0$ **do**

**17** $\quad\quad$ $(\widehat{X}_t, \widehat{y}_t) \leftarrow$ DataGeneration($n_t$, $M_t$, $\mu_t$, $\pi_t$, $C_t$, $\widehat{W}_{t+1}^{\lambda_{t+1}}$);

**18** $\quad\quad$ $\widehat{W}_t^{\lambda_t} \leftarrow$ Regression($(\widehat{X}_t, \widehat{y}_t)$, $\Theta_t$); $\qquad$ `// KRR/FALKON on (X̂_t, ŷ_t)`

**19** $\quad$ **end**

**20** $\quad$ **return** $\widehat{W}_0^{\lambda_0}$

---

## 4    ERROR ANALYSIS AND BACKWARD PROPAGATION

In this section, our primary goal is to provide theoretical guarantees for our estimator $\widehat{W}_t^{\lambda_t}$ and to study how the error propagates backward in time from $T$ to $0$. In particular, we are interested in analyzing the rate of convergence of $\widehat{W}_t^{\lambda_t}$ to the target value function $V_t$ in some norm, as a function of the sample sizes $n_t$ and $M_t$. A natural choice is to bound

$$\mathcal{E}_t = \left\| \widehat{W}_t^{\lambda_t} - V_t \right\|_{L_{\mu_t}^2}^2. \tag{22}$$

### 4.1    ERROR DECOMPOSITION

To do so, we split the total error into three components:

$$\mathcal{E}_t \lesssim \left\| \widehat{W}_t^{\lambda_t} - \widetilde{\mathcal{T}}_t \widehat{W}_{t+1}^{\lambda_{t+1}} \right\|_{L_{\mu_t}^2}^2 + \left\| \widetilde{\mathcal{T}}_t \widehat{W}_{t+1}^{\lambda_{t+1}} - \mathcal{T}_t \widehat{W}_{t+1}^{\lambda_{t+1}} \right\|_{L_{\mu_t}^2}^2 + \left\| \mathcal{T}_t \widehat{W}_{t+1}^{\lambda_{t+1}} - \mathcal{T}_t V_{t+1} \right\|_{L_{\mu_t}^2}^2. \tag{23}$$

Notice that the rigorous mathematical formulation established above ensures the well-definedness of all terms, allowing us to meaningfully leverage this structure to decompose the total error. The resulting splitting is a key advantage of our framework: the error separates into three distinct components, each associated with a different approximation step. Because the bound is fully modular, each term can be improved independently without affecting the rest of the analysis. Term I depends only on the regression method—while we use kernel ridge regression for its clear statistical guarantees, other function approximators (e.g., neural networks) could be substituted. Term II isolates the sampling error, so Monte Carlo can be replaced by more refined schemes such as quadrature or variance-reduced estimators. Term III captures the intrinsic propagation induced by the dynamics and is unaffected by choices in the first two terms. This flexible structure allows us to work on each component independently while preserving the overall analysis.

**Term I: Regression Error.**    The first term is the standard machine learning error due to the fact that our estimator minimizes the empirical risk in Eq. 16, based only on a finite sample $\{(x_i, y_i)\}_{i=1}^{n_t}$. Our target is the regression function $W_t^* = \widetilde{\mathcal{T}}_t \widehat{W}_{t+1}^{\lambda_{t+1}}$, as defined in Eq. 17. Term I then corresponds to the so-called excess risk of $\widehat{W}_t^{\lambda_t}$:

$$\mathcal{R}(\widehat{W}_t^{\lambda_t}) - \mathcal{R}(W_t^*) := \mathbb{E}\left[ (Y - \widehat{W}_t^{\lambda_t}(X))^2 - (Y - W_t^*(X))^2 \right] = \left\| \widehat{W}_t^{\lambda_t} - W_t^* \right\|_{L_{\mu_t}^2}^2, \tag{24}$$

(see Caponnetto & De Vito (2007)), where $\mathcal{R}(\widehat{W}_t^{\lambda_t})$ is the risk of $\widehat{W}_t^{\lambda_t}$ and $\mathcal{R}(W_t^*) = \mathcal{R}(\widetilde{\mathcal{T}}_t \widehat{W}_{t+1}^{\lambda_{t+1}})$. It represents the expected error of our estimator on new data compared to the regression function.

We introduce the following regularity assumption, commonly referred to as the *source condition*.

**Assumption 3** (Source Condition). *There exists $\beta_t \in (0, 1]$ such that $W_t^* \in L_k^{\beta_t/2}(L_{\mu_t}^2)$, where $L_k : L_{\mu_t}^2 \to L_{\mu_t}^2$ is the integral operator associated with the kernel $k$.*

Assumption 3 and equivalent formulations (e.g., Assumption 4 in Rudi et al. (2015a)) are standard in the literature (Smale & Zhou, 2007; Caponnetto & De Vito, 2007). The parameter $\beta_t$ quantifies the smoothness of the target function $W_t^*$ and how well it can be approximated by elements in $\mathcal{H}_k$. When $\beta_t = 1$, we are in the *well-specified* setting, i.e., $W_t^* \in \mathcal{H}_k$. Our main focus, however, is on the *misspecified* setting with $\beta_t < 1$, where $W_t^* \notin \mathcal{H}_k$.

Under the square loss, Assumption 3 is directly related to the approximation error, as shown in Smale & Zhou (2003); Steinwart et al. (2009). Using a result from Corollary 6 in Steinwart et al. (2009) (see Appendix B.1), we obtain the following upper bound in terms of $n_t$: choosing $\lambda_t \sim n_t^{-\frac{1}{\beta_t+1}}$, with high probability,

$$\left\| \widehat{W}_t^{\lambda_t} - \widetilde{\mathcal{T}}_t \widehat{W}_{t+1}^{\lambda_{t+1}} \right\|_{L_{\mu_t}^2}^2 \lesssim n_t^{-\frac{\beta_t}{\beta_t+1}}. \tag{25}$$

We refer to Appendix B.1 for further details. Note that the above rate can be made faster by assuming some polynomial (or even exponential) decay of the spectrum of the integral operator $L_k$. This is deeply connected to the well-known *capacity assumption*, which for simplicity is not assumed here in the main text. Further details and the resulting faster rate can be found in Appendix B.1.

**Term II: Monte Carlo Error.** The second term accounts for the Monte Carlo error introduced when approximating the unknown expectation in $\mathcal{T}_t$, as discussed in Section 3.

Using the definitions of $\mathcal{T}_t$ and $\widetilde{\mathcal{T}_t}$ from Eqs. 6 and 13, together with Lemma 1 in Appendix A, and denoting $\mathcal{F}_t^x = \{z \mapsto \widehat{W}_{t+1}^{\lambda_{t+1}}(\pi_t(x, u, z)) : u \in \mathcal{U}_t\}$, we obtain that, with high probability,

$$\left\|\widetilde{\mathcal{T}_t}\widehat{W}_{t+1}^{\lambda_{t+1}} - \mathcal{T}_t\widehat{W}_{t+1}^{\lambda_{t+1}}\right\|_{L_{\mu_t}^2}^2 \leqslant \left\|\sup_{f \in \mathcal{F}_t^x}\left|\frac{1}{M_t}\sum_{j=1}^{M_t} f(z_j) - \mathbb{E}[f(Z_{t+1})]\right|\right\|_{L_{\mu_t}^2}^2 \lesssim \left\|\mathbb{E}\widehat{\mathcal{R}}(\mathcal{F}_t^x) + \sqrt{\frac{1}{M_t}}\right\|_{L_{\mu_t}^2}^2$$

where the last inequality follows from the boundedness of $\widehat{W}_{t+1}^{\lambda_{t+1}}$ and an application of Boucheron et al. (2005, Theorem 3.2), while $\widehat{\mathcal{R}}(\mathcal{F}_t^x)$ denotes the well-known empirical Rademacher complexity of $\mathcal{F}_t^x$ (see definition in Appendix B.2). Bounding such complexities is a classical problem in statistical learning theory (Bartlett & Mendelson, 2002). In our setting, we focus on two relevant cases: (i) finite classes, as in American options where the control set is binary ($\mathcal{U}_t = \{0, 1\}$), and (ii) Lipschitz transitions $\pi_t$, which are typical in financial models once the state space is a compact set (see the discussion about truncation in previous section). Using results from Massart (2000); Bartlett & Mendelson (2002) (see Appendix B.2), we obtain for both cases

$$\mathbb{E}\widehat{\mathcal{R}}(\mathcal{F}_t^x) \lesssim \sqrt{1/M_t}. \tag{26}$$

**Term III: Propagation Error.** This term captures the error inherited from the previous step $t + 1$. By Lemma 2 in Appendix A, we have:

$$\left\|\mathcal{T}_t\widehat{W}_{t+1}^{\lambda_{t+1}} - \mathcal{T}_t V_{t+1}\right\|_{L_{\mu_t}^2}^2 \leqslant c_P\left\|\widehat{W}_{t+1}^{\lambda_{t+1}} - V_{t+1}\right\|_{L_{\mu_t}^2}^2 = c_P\mathcal{E}_{t+1}.$$

**Final Bound.** Putting everything together, we obtain the following result.

**Theorem 1** (Error Backpropagation). *Under Assumptions 1, 2, 3, and provided that condition 26 holds, with the choice $\lambda_t \sim n_t^{-\frac{1}{\beta_t+1}}$ and $M_t \sim n_t^{\frac{\beta_t}{\beta_t+1}}$, we have with high probability:*

$$\mathcal{E}_t = \left\|\widehat{W}_t^{\lambda_t} - V_t\right\|_{L_{\mu_t}^2}^2 \lesssim \left(\frac{1}{n_t}\right)^{\frac{\beta_t}{\beta_t+1}} + c_P\mathcal{E}_{t+1}, \tag{27}$$

*for $t \in \{0, \ldots, T-1\}$. Furthermore,*

$$\mathcal{E}_0 = \left\|\widehat{W}_0^{\lambda_0} - V_0\right\|_{L_{\mu_0}^2}^2 \lesssim \sum_{t=0}^{T-1} c_P^t\left(\frac{1}{n_t}\right)^{\frac{\beta_t}{\beta_t+1}}. \tag{28}$$

Note that, as desirable, the error vanishes as $n_t \to \infty$ for all $t$. In the non-asymptotic regime, the convergence rate depends on the smoothness parameters $\{\beta_t\}_t$, which reflect the level of misspecification of the problem. Although the expectation operator $\widehat{P}_t^u$ may act as a smoothing operator, the supremum in the Bellman operator prevents us from guaranteeing a smoothing effect through time. As a result, the problem generally remains misspecified throughout the backward recursion. Note also that the constant $c_P$ in Assumption 1 plays a key role in controlling the resulting error propagation. When $c_P < 1$, as in our option pricing setting (see Example 3 below), the recursion becomes contractive, so errors are damped rather than amplified, making convergence faster and more stable.

**Example 3** (American Options (cont.)). *Returning to our application to American option pricing in Example 1, we now adapt Theorem 1 to this setting. Note that $W_T^* = V_T = \Psi$ is typically non-smooth for common payoff functions, see Eq. 9 or Fig. 1-2 in Appendix C). As mentioned above, this places us in the misspecified case, where the smoothness parameters $\{\beta_t\}_t$ can be small, while it is not clear if the specification eventually improves throughout the recursion. From Eq. 12, the Bellman operator $\mathcal{T}_t : L_{\mu_{t+1}}^2 \to L_{\mu_t}^2$ takes the form:*

$$\mathcal{T}_t g = \max\left(C_t, \ e^{-r} Q_t g\right). \tag{29}$$

*We now verify that the assumptions required by Theorem 1 are satisfied. First condition in Eq. 8 in Assumption 1 is straightforward since $U_t = \{0, 1\}$ and $F_t$ is defined as in Eq. 10:*

Table 1: Results for a Geometric basket Put option, see Table 1 in Goudenege et al. (2020).

| | **KRR-DP** | | | **GPR-Tree** | | **GPR-EI** | | **Ekvall** | **Benchmark** |
|---|---|---|---|---|---|---|---|---|---|
| $d$ | Price | 95% CI | Time | Price | Time | Price | Time | Price | Price |
| 2 | 4.63 | [4.58, 4.68] | 2s | 4.61 | 22s | 4.57 | 26s | 4.62 | 4.62 |
| 5 | 3.46 | [3.42, 3.50] | 3s | 3.44 | 23s | 3.41 | 27s | 3.44 | 3.45 |
| 10 | 2.98 | [2.94, 3.03] | 4s | 2.93 | 60s | 2.93 | 30s | 2.90 | 2.97 |
| 20 | 2.70 | [2.68, 2.72] | 21s | 2.72 | 49609s | 2.63 | 29s | 2.70 | 2.70 |
| 40 | 2.55 | [2.53, 2.57] | 87s | / | / | 2.53 | 38s | 2.57 | 2.56 |

Table 2: Results for a Max-Call option, see Table 3 in Goudenege et al. (2020).

| | **KRR-DP** | | | **GPR-Tree** | | **GPR-EI** | | **GPR-MC** | **Ekvall** |
|---|---|---|---|---|---|---|---|---|---|
| $d$ | Price | 95% CI | Time | Price | Time | Price | Time | Price | Price |
| 2 | 16.93 | [16.86, 17.00] | 5s | 16.93 | 20s | 16.82 | 28s | 16.86 | 16.86 |
| 5 | 27.16 | [26.98, 27.33] | 5s | 27.19 | 26s | 26.95 | 27s | 27.20 | 27.20 |
| 10 | 35.14 | [34.94, 35.35] | 6s | 35.08 | 106s | 34.84 | 29s | 35.17 | / |
| 20 | 42.62 | [42.30, 42.93] | 27s | 43.00 | 51090s | 42.62 | 35s | 42.76 | / |
| 40 | 50.44 | [50.01, 50.87] | 118s | / | / | 49.53 | 41s | 50.70 | / |

ess sup$_{u \in \{0,1\}} |F_t(\cdot, u)| = F_t(\cdot, 0) = C_t$. *We let $c_F$ be the squared $L^2_{\mu_t}$-norm of $C_t$, which is assumed to be finite. Moreover, since $Q_t$ is a Markov transition kernel, it defines a non-expansive operator:*

$$\|Q_t g\|^2_{L^2_{\mu_t}} = \int_{\mathcal{X}_t} \left( \int_{\mathcal{X}_{t+1}} g(x')Q(x, dx') \right)^2 d\mu_t \leqslant \int_{\mathcal{X}_{t+1}} g(x')^2 \int_{\mathcal{X}_t} Q(x, dx') d\mu_t \leqslant \|g\|^2_{L^2_{\mu_{t+1}}},$$

*where we used Jensen's inequality and Fubini's theorem. Therefore, condition 8 in Assumption 1 is also satisfied. We can now bound the Bellman operator $\mathcal{T}_t$:*

$$\|\mathcal{T}_t g\|_{L^2_{\mu_t}} = \left\| \max \left( C_t, \, e^{-r} Q_t g \right) \right\|_{L^2_{\mu_t}} \leqslant c_F + e^{-r} \|g\|_{L^2_{\mu_{t+1}}} \tag{30}$$

*Then, Assumption 1 is satisfied with $c_P = e^{-2r}$. Note also that $c_P < 1$ in the common case of a strictly positive risk-free interest rate $r$.*

**Corollary 1** (American Option Pricing)**.** *From Theorem 1, in the setting described in Example 1, 2 and 3, and following Algorithm 1, we have with high probability:*

$$\left\| \widehat{W}_0^{\lambda_0} - V_0 \right\|^2_{L^2_{\mu_0}} \lesssim \sum_{t=0}^{T-1} e^{-rt} \left( \frac{1}{n_t} \right)^{\frac{\beta_t}{\beta_t + 1}}. \tag{31}$$

*Notice that, in the common case of a positive interest rate $e^{-rt} < 1$ (i.e. $c_P < 1$), the Bellman operator is contractive in this setting. This contraction plays a crucial role in the corollary: it damps the propagated errors across time steps, thereby facilitating convergence of the overall approximation.*

## 5 SIMULATIONS

In this section, we present a basic implementation of KRR-DP Algorithm 1 and conduct an initial evaluation of the effectiveness of the proposed method. More comprehensive experiments and optimized implementations will be the subject of future work.

We primarily compare our results with the numerical benchmarks reported in Goudenege et al. (2020). Specifically, we replicate the results in their Table 1 and Table 3, which correspond to pricing a geometric basket put option and a max-call option, respectively. Note that a theoretical benchmark exists for geometric basket put options. The parameters are set reproducing the ones in Goudenege et al. (2020) and are the following: time horizon $T = 9$, initial state $X_0^i = 100$ for $1 \leqslant i \leqslant d$, strike price $K = 100$, free-risk interest rate $r = 0.05$, volatility $\sigma_i = 0.2$ for $1 \leqslant i \leqslant d$,

and correlation $\rho_{ij} = 0.2$ for $1 \leqslant i \neq j \leqslant d$.

As regards the KRR solver, we employ the efficient FALKON algorithm (Meanti et al., 2020), which uses Nyström-based random projections (Williams & Seeger, 2000) to achieve substantial computational savings while retaining optimal statistical performance (Rudi et al., 2015b; Della Vecchia et al., 2021; 2024). Importantly, FALKON is fully compatible with our theoretical analysis: its Nyström approximation enjoys provably optimal excess-risk guarantees (given a sufficient number of centers), and thus can directly replace the full KRR estimator in Term I of our error decomposition (see Section 6 in Della Vecchia et al. (2024)). Thanks to the modular structure of our bounds, this substitution preserves the convergence rate in Eq. (25), does not affect the other terms, and leaves the overall propagation rate in Eq. (28) unchanged—while significantly reducing computational cost. This illustrates a key advantage of the neat and transparent structure of our framework: each source of error can be improved independently (e.g., by refining Monte Carlo or replacing it with more efficient quadrature) without altering the overall analysis. A description of the involved methods and further details on our simulations are given in Appendix C.

These preliminary results suggest that our method performs competitively in this setting with the considered benchmarks, offering a favorable trade-off between accuracy and computational efficiency. As the dimension increases, however, the runtime grows noticeably, with the Monte Carlo component (Term II) emerging as the main computational bottleneck. The study and incorporation of alternatives to standard Monte Carlo capable of reducing the required number of samples $M$ while preserving statistical guarantees (such as the above-mentioned quadrature rules) will be the basis for a competitive large-scale implementation, and constitute the natural direction for follow-up work.

## 6 CONCLUSIONS AND FUTURE WORK

In this work, we addressed stochastic optimal control problems in discrete time and introduced a kernel-based regression framework for their solution. Our approach combines backward recursion via empirical Bellman operators with Monte Carlo simulation and regularized learning techniques to construct data-driven approximations of the value function. The framework is supported by rigorous theoretical guarantees, including explicit error bounds.

Several promising directions remain open for future work. First, we plan to extend the preliminary simulations presented above into a more comprehensive experimental study, incorporating real-world datasets and more complex models. In particular, our framework can naturally be adapted to other non-standard applications in economics, such as partial equilibrium, optimal consumption, or goal-based investing.

In parallel, there is significant room to improve computational efficiency, especially in high-dimensional settings, by developing a competitive implementation of the algorithm. While the regression step can be accelerated, as in our simulations, using established scalability techniques that preserve the statistical guarantees of KRR (e.g., sketching, random features, Nyström approximations), the main bottleneck in our pipeline appears to be the Monte Carlo data-generation step. Reducing the number $M$ of simulated points is critical for speeding up the DATAGENERATION function in Algorithm 1. A possible approach is to replace standard Monte Carlo sampling with more sophisticated quadrature schemes (e.g., monomial rules), which may significantly reduce computational cost while maintaining accuracy.

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

# A    AUXILIARY LEMMAS

In this section, we prove a number of technical results that are instrumental for establishing the theoretical properties of our Bellman recursion in $L^2_{\mu_t}$ spaces. In particular, we aim to verify that the Bellman operator $\mathcal{T}_t$ is well defined and Lipschitz continuous under mild assumptions. These properties are essential for proving stability and convergence of our value function approximations.

We begin with a useful lemma on the behavior of essential suprema, which allows us to control expressions of the form $\operatorname{ess\,sup}_{u \in U_t} \{F_t(\cdot, u) + P^u_t g\}$ arising in the Bellman operator.

**Lemma 1.** *Let $\{Y_a\}_{a \in A}$ and $\{Z_a\}_{a \in A}$ be two collections of random variables indexed by a parameter set $A$, such that $\operatorname{ess\,sup}_{a \in A} |Y_a| < \infty$ and $\operatorname{ess\,sup}_{a \in A} |Z_a| < \infty$ almost surely. Then the following inequalities hold almost surely:*

$$\left| \operatorname*{ess\,sup}_{a \in A} (Y_a + Z_a) \right| \leqslant \operatorname*{ess\,sup}_{a \in A} |Y_a| + \operatorname*{ess\,sup}_{a \in A} |Z_a|, \tag{32}$$

$$\left| \operatorname*{ess\,sup}_{a \in A} Y_a - \operatorname*{ess\,sup}_{a \in A} Z_a \right| \leqslant \operatorname*{ess\,sup}_{a \in A} |Y_a - Z_a|. \tag{33}$$

*Proof.* The first bound follows from the general inequality $|\operatorname{ess\,sup}_{a \in A} Y_a| \leqslant \operatorname{ess\,sup}_{a \in A} |Y_a|$. For the second inequality, we exploit the invariance under translations: the statement holds if we replace $Y_a$ and $Z_a$ by $Y_a + C$ and $Z_a + C$, for any random variable $C$. Choosing $C = \max\{\operatorname{ess\,sup}_{a \in A}(-Y_a), \operatorname{ess\,sup}_{a \in A}(-Z_a)\}$, we can assume without loss of generality that $Y_a, Z_a \geqslant 0$. Then we obtain from Eq. 32 that $\operatorname{ess\,sup}_{a \in A} Y_a \leqslant \operatorname{ess\,sup}_{a \in A} |Y_a - Z_a| + \operatorname{ess\,sup}_{a \in A} Z_a$, and same for $Y_a$ and $Z_a$ exchanged, which proves Eq. 33. $\qquad\square$

With this result in hand, we now analyze the properties of the Bellman operator $\mathcal{T}_t$ as defined in Eq. 6. The following lemma shows that, under suitable assumptions, $\mathcal{T}_t$ maps $L^2_{\mu_{t+1}}$ to $L^2_{\mu_t}$ in a controlled way and satisfies a global Lipschitz bound.

**Lemma 2.** *Under conditions 8 in Assumption 1, the Bellman operator $\mathcal{T}_t$ defines a Lipschitz continuous map satisfying:*

$$\|\mathcal{T}_t g\|_{L^2_{\mu_t}} \leqslant c_F + c_P^{1/2} \|g\|_{L^2_{\mu_{t+1}}}, \tag{34}$$

$$\|\mathcal{T}_t g - \mathcal{T}_t f\|^2_{L^2_{\mu_t}} \leqslant c_P \|g - f\|^2_{L^2_{\mu_{t+1}}}, \tag{35}$$

*for all $g, f \in L^2_{\mu_{t+1}}$ and $t = 0, \dots, T - 1$.*

*Proof.* We begin by bounding the operator norm:

$$\|\mathcal{T}_t g\|_{L^2_{\mu_t}} = \left\| \operatorname*{ess\,sup}_{u \in U_t} \{F_t(\cdot, u) + P^u_t g\} \right\|_{L^2_{\mu_t}}$$

$$\leqslant \left\| \operatorname*{ess\,sup}_{u \in U_t} |F_t(\cdot, u)| + \operatorname*{ess\,sup}_{u \in U_t} |P^u_t g| \right\|_{L^2_{\mu_t}}$$

$$\leqslant \left\| \operatorname*{ess\,sup}_{u \in U_t} |F_t(\cdot, u)| \right\|_{L^2_{\mu_t}} + \left\| \operatorname*{ess\,sup}_{u \in U_t} |P^u_t g| \right\|_{L^2_{\mu_t}}$$

$$\leqslant c_F + c_P^{1/2} \|g\|_{L^2_{\mu_{t+1}}},$$

where we used Lemma 1 and Assumption 1. For the Lipschitz property, we compute:

$$\|\mathcal{T}_t g - \mathcal{T}_t f\|_{L^2_{\mu_t}} = \left\| \operatorname*{ess\,sup}_{u \in U_t} \{F_t(\cdot, u) + P^u_t g\} - \operatorname*{ess\,sup}_{u \in U_t} \{F_t(\cdot, u) + P^u_t f\} \right\|_{L^2_{\mu_t}}$$

$$\leqslant \left\| \operatorname*{ess\,sup}_{u \in U_t} |P^u_t (g - f)| \right\|_{L^2_{\mu_t}}$$

$$\leqslant c_P^{1/2} \|g - f\|_{L^2_{\mu_{t+1}}},$$

again applying Lemma 1 and that $P^u_t$ is a linear operator. $\qquad\square$

# B TECHNICAL DETAILS ON SECTION 4

## B.1 TERM I

In this section, we give further details about the analysis of our learning-based approximation scheme in Section 4.

We start with the optimal learning rates established for regularized empirical risk minimization in RKHS. The following theorem is taken from Steinwart et al. (2009).

**Theorem** (Steinwart et al. (2009, Theorem 1)). *Let $k$ be a bounded measurable kernel on $X$ with $\|k\|_\infty = 1$ and separable RKHS $\mathcal{H}$. Let*

$$A_q(\lambda) := \inf_{f \in \mathcal{H}} \left( \lambda \|f\|_{\mathcal{H}}^q + \mathcal{R}(f) - \mathcal{R}^* \right). \tag{36}$$

*Moreover, let $P$ be a distribution on $X \times [-B, B]$, where $B > 0$ is some constant. For $\nu = P_X$ assume that the extended sequence of eigenvalues of the integral operator satisfies*

$$\mu_i (L_k) \leqslant ai^{-\frac{1}{p}}, \quad i \geqslant 1, \tag{37}$$

*where $a \geqslant 16M^4$ and $p \in (0,1)$. Assume further that there exist constants $C \geqslant 1$ and $s \in (0,1]$ such that*

$$\|f\|_\infty \leqslant C\|f\|_{\mathcal{H}}^s \cdot \|f\|_{L_2(P_X)}^{1-s} \tag{38}$$

*for all $f \in \mathcal{H}$. Then, for all $q \geqslant 1$, there exists a constant $c_{p,q}$ depending only on $p$ and $q$ such that for all $\lambda \in (0,1]$, $\tau > 0$, and $n \geqslant 1$, with probability at least $1 - 3e^{-\tau}$*

$$\mathcal{R}\left(\widehat{f}_\lambda\right) - \mathcal{R}^* \leqslant 9A_q(\lambda) + c_{p,q} \left( \frac{a^{pq}B^{2q}}{\lambda^{2p}n^q} \right)^{\frac{1}{q-2p+pq}} + \frac{120C^2B^{2-2s}\tau}{n} \left( \frac{A_q(\lambda)}{\lambda} \right)^{\frac{2s}{q}} + \frac{3516B^2\tau}{n} \tag{39}$$

*with $\mathcal{R}^* := \mathcal{R}(f^*)$ the risk of the Bayes function $f^* \in L^2(P_X)$ and $\widehat{f}_\lambda$ the data dependent estimator from ERM algorithm.*

Note that Eq 37 is exactly the condition mentioned under Eq. 25. We give here more details on the connection with the *capacity assumption*. Before defining it, we define the so-called *effective dimension* (Zhang, 2005; Caponnetto & De Vito, 2007), for $\alpha > 0$, as

$$d_\alpha = \text{Tr}((L_k + \alpha I)^{-1}L_k) = \sum_j \frac{\sigma_j}{\sigma_j + \alpha} \tag{40}$$

where $(\sigma_j)_j$ are the strictly positive eigenvalues of $L_k$, with eigenvalues counted with respect to their multiplicity and ordered in a non-increasing way, and $(u_j)$ is the corresponding family of eigenvectors.

**Assumption 4** (Capacity Assumption). *There exist constants $p \geqslant 1$ and $Q > 0$ such that, for all $\alpha \in (0,1]$*

$$d_\alpha \leqslant Q\alpha^{-1/p}.$$

This assumption, standard in statistical learning theory (see Caponnetto & De Vito, 2007; Smale & Zhou, 2007), is often referred to as a capacity condition, as it quantifies the effective size of the RKHS via the decay of the eigenvalues of the integral operator $L_k$ (see Proposition 1 and 2 below). Note that the case $p = 1$ corresponds to no spectral assumption (i.e. the weakest possible capacity control), which is the setting we adopt in the main text.

The following two results provide a tight bound on the effective dimension under the assumption of a polynomial decay or an exponential decay of the eigenvalues $\sigma_j$ of $L_k$. Since the covariance operator $\Sigma$ and the integral operator $L_k$ share the same eigenvalues, we equivalently report known proofs for $\Sigma$ in the following.

**Proposition 1** (Polynomial eigenvalues decay Caponnetto & De Vito (2007, Proposition 3)). *If for some $\gamma \in \mathbb{R}^+$ and $1 < p < +\infty$*

$$\sigma_i \leqslant \gamma i^{-p}$$

*then*

$$d_\alpha \leqslant \gamma \frac{p}{p-1} \alpha^{-1/p} \tag{41}$$

*Proof.* Since the function $\sigma/(\sigma + \alpha)$ is increasing in $\sigma$ and using the spectral theorem $\Sigma = UDU^*$ combined with the fact that $\mathrm{Tr}(UDU^*) = \mathrm{Tr}(U(U^*D)) = \mathrm{Tr}D$

$$d_\alpha = \mathrm{Tr}(\Sigma(\Sigma + \alpha I)^{-1}) = \sum_{i=1}^{\infty} \frac{\sigma_i}{\sigma_i + \alpha} \leqslant \sum_{i=1}^{\infty} \frac{\gamma}{\gamma + i^p \alpha} \tag{42}$$

The function $\gamma/(\gamma + x^p \alpha)$ is positive and decreasing, so

$$\begin{aligned} d_\alpha &\leqslant \int_0^\infty \frac{\gamma}{\gamma + x^p \alpha} dx \\ &= \alpha^{-1/p} \int_0^\infty \frac{\gamma}{\gamma + \tau^p} d\tau \\ &\leqslant \gamma \frac{p}{p-1} \alpha^{-1/p} \end{aligned} \tag{43}$$

since $\int_0^\infty (\gamma + \tau^p)^{-1} \leqslant p/(p-1)$. $\qquad\square$

A similar result, leading to even faster rates, can be obtained assuming an exponential decay.

**Proposition 2** (Exponential eigenvalues decay Della Vecchia et al. (2024, Proposition 3)). *If for some $\gamma, p \in \mathbb{R}^+ \sigma_i \leqslant \gamma e^{-pi}$ then*

$$d_\alpha \leqslant \frac{\log(1 + \gamma/\alpha)}{p} \tag{44}$$

*Proof.*

$$d_\alpha = \sum_{i=1}^{\infty} \frac{\sigma_i}{\sigma_i + \alpha} = \sum_{i=1}^{\infty} \frac{1}{1 + \alpha/\sigma_i} \leqslant \sum_{i=1}^{\infty} \frac{1}{1 + \alpha' e^{pi}} \leqslant \int_0^{+\infty} \frac{1}{1 + \alpha' e^{px}} dx \tag{45}$$

where $\alpha' = \alpha/\gamma$. Using the change of variables $t = e^{px}$ we get

$$\begin{aligned} (45) &= \frac{1}{p} \int_1^{+\infty} \frac{1}{1 + \alpha' t} \frac{1}{t} dt = \frac{1}{p} \int_1^{+\infty} \left[ \frac{1}{t} - \frac{\alpha'}{1 + \alpha' t} \right] dt = \frac{1}{p} \left[ \log t - \log(1 + \alpha' t) \right]_1^{+\infty} \\ &= \frac{1}{p} \left[ \log\left( \frac{t}{1 + \alpha' t} \right) \right]_1^{+\infty} = \frac{1}{p} \left[ \log(1/\alpha') + \log(1 + \alpha') \right] \end{aligned} \tag{46}$$

So we finally obtain

$$d_\alpha \leqslant \frac{1}{p} \left[ \log(\gamma/\alpha) + \log(1 + \alpha/\gamma) \right] = \frac{\log(1 + \gamma/\alpha)}{p} \tag{47}$$

$\qquad\square$

Specializing this result to ridge regression, and under an additional approximation condition on the learning target, we obtain a more explicit convergence rate in terms of the sample size.

**Corollary** (Steinwart et al. (2009, Corollary 6)). *Assume $s = p = 1$, $q = 2$, and suppose the 2-approximation error function satisfies*

$$A_2(\lambda) \leqslant c\lambda^\beta, \quad \lambda > 0 \tag{48}$$

*for some constants $c > 0$ and $\beta > 0$. Define a sequence of regularization parameters $\lambda := n^{-\frac{1}{\beta+1}}$. Then there exists a constant $K \geqslant 1$ depending only on $a$, $B$, and $c$, such that for all $\tau \geqslant 1$ and $n \geqslant 1$,*

$$\mathcal{R}\left( \widehat{f}_\lambda \right) - \mathcal{R}(f^*) \leqslant K\tau n^{-\frac{\beta}{\beta+1}} \tag{49}$$

*with probability at least $1 - 3e^{-\tau n^{\frac{\beta}{\beta+1}}}$.*

This is the result reported in Theorem 1, given that source condition in Assumption 3 implies condition in Eq.48 as shown in Smale & Zhou (2003).

## B.2 TERM II

We start by defining the empirical Rademacher complexity:

$$\widehat{\mathcal{R}}(\mathcal{F}_t^x) := \mathbb{E}_\sigma \sup_{f \in \mathcal{F}_t^x} \left| \frac{1}{M_t} \sum_{i=1}^{M_t} \sigma_i f(z_i) \right|, \tag{50}$$

with $\sigma_1, \ldots, \sigma_{M_t}$ independent Rademacher variables, i.e. $\mathbb{P}(\sigma_i = 1) = \mathbb{P}(\sigma_i = -1) = 1/2$.

To control the empirical approximation error uniformly over a function class, we rely on the following concentration inequality due to Boucheron et al. (2005).

**Lemma** (Boucheron et al. (2005, Theorem 3.2)). *Let $X_1, \ldots, X_n$ be i.i.d. random variables in a set $\mathcal{X}$ and let $\mathcal{F}$ be a class of functions $\mathcal{X} \to [-1, 1]$. Then, with probability at least $1 - \delta$,*

$$\sup_{f \in \mathcal{F}} \left| \mathbb{E}f(X) - \frac{1}{n} \sum_{i=1}^n f(X_i) \right| \leqslant 2\mathbb{E}\widehat{\mathcal{R}}(\mathcal{F}(X_1^n)) + \sqrt{\frac{2 \log \frac{1}{\delta}}{n}}, \tag{51}$$

*with*

$$\widehat{\mathcal{R}}(A) = \mathbb{E} \sup_{a \in A} \frac{1}{n} \left| \sum_{i=1}^n \sigma_i a_i \right|, \tag{52}$$

*where $A \subset \mathbb{R}^n$ and $\mathcal{F}(x_1^n)$ is the class of vectors $(f(x_1), \ldots, f(x_n))$ for $f \in \mathcal{F}$.*

*We also have:*

$$\sup_{f \in \mathcal{F}} \left| \mathbb{E}f(X) - \frac{1}{n} \sum_{i=1}^n f(X_i) \right| \leqslant 2\widehat{\mathcal{R}}(\mathcal{F}(X_1^n)) + \sqrt{\frac{2 \log \frac{2}{\delta}}{n}}. \tag{53}$$

There are several well-studied cases in which the Rademacher complexity can be upper bounded. We highlight two such cases that are particularly relevant for the financial applications of interest here.

- Using Massart's Lemma (Massart, 2000): if $\mathcal{F}_t^x$ is finite, i.e., $\mathcal{F}_t^x = \{f_1, \ldots, f_K\}$, then

$$\mathbb{E}\widehat{\mathcal{R}}(\mathcal{F}_t^x) \lesssim \sqrt{\frac{\log K}{M_t}}. \tag{54}$$

  This result is particularly relevant for our application to American options, as the control set $\mathcal{U}_t = \{0, 1\}$ is finite at each time step $t$.

- Using Talagrand's Contraction Lemma (Ledoux & Talagrand, 1991): if $\mathcal{F}_t^x$ is not finite, $\widehat{W}_{t+1}^{\lambda_{t+1}}$ is $L_W$-Lipschitz, and we define $\Pi_t^x := \{z \mapsto \pi_t(x, u, z) : u \in \mathcal{U}_t\}$, then the composition class $\mathcal{F}_t^x = \widehat{W}_{t+1}^{\lambda_{t+1}} \circ \Pi_t^x$ satisfies

$$\widehat{\mathcal{R}}(\mathcal{F}_t^x) \leqslant L_W \cdot \widehat{\mathcal{R}}(\Pi_t^x). \tag{55}$$

  Assuming that $\pi_t(x, u, z)$ is $L_\pi$-Lipschitz in $u$ and applying standard covering number arguments we obtain

$$\mathbb{E}\widehat{\mathcal{R}}(\mathcal{F}_t^x) \lesssim \frac{L_W \cdot L_\pi}{\sqrt{M_t}}. \tag{56}$$

  This can be useful in the continuous control case, e.g., $\mathcal{U}_t \subset [0, 1]$, as the class $\Pi_t^x$ is no longer finite.

We report the two above mentioned results.

**Lemma** (Massart's Lemma (Massart, 2000), (Shalev-Shwartz & Ben-David, 2014, Lemma 26.8)). *Let $\mathcal{F} = \{f_1, \ldots, f_K\}$ be a finite class of functions satisfying $\|f\|_\infty \leqslant b$ for all $f \in \mathcal{F}$. Then,*

$$\widehat{\mathcal{R}}(\mathcal{F}) \leqslant b\sqrt{\frac{2 \log K}{n}}. \tag{57}$$

**Lemma** (Contraction Inequality (Bartlett & Mendelson, 2002, Thm. 12), (Ledoux & Talagrand, 1991, Cor. 3.17)). *Let $\mathcal{F} \subset \mathbb{R}^{\mathcal{Z}}$ be a class of real-valued functions, and let $\phi_1, \ldots, \phi_n : \mathbb{R} \to \mathbb{R}$ be L-Lipschitz functions. Let $S = \{z_1, \ldots, z_n\} \subset \mathcal{Z}$ be a fixed sample. Then*

$$\mathbb{E}_{\sigma} \left[ \sup_{f \in \mathcal{F}} \frac{1}{n} \sum_{i=1}^{n} \sigma_i \phi_i(f(z_i)) \right] \leqslant L \cdot \mathbb{E}_{\sigma} \left[ \sup_{f \in \mathcal{F}} \frac{1}{n} \sum_{i=1}^{n} \sigma_i f(z_i) \right], \tag{58}$$

*where $\sigma_1, \ldots, \sigma_n$ are independent Rademacher random variables.*

### B.3 FINAL BOUND

Given the above upper bounds on the three terms in Eq. 23, and choosing $\lambda \sim n^{-\frac{1}{\beta_t+1}}$, we have with high probability

$$\mathcal{E}_t \lesssim \left( \frac{1}{n_t} \right)^{\frac{\beta_t}{\beta_t+1}} + \frac{1}{M_t} + c_P \mathcal{E}_{t+1}. \tag{59}$$

Selecting $M_t \sim n_t^{\frac{\beta_t}{\beta_t+1}}$ gives the result in Theorem 1.

## C NUMERICAL SIMULATIONS

Firstly, we briefly describe the benchmark methods used for comparison in Tables 1 and 2, following Goudenege et al. (2020).

**GPR-Tree.** This method combines Gaussian Process Regression (GPR) with a tree-based exercise strategy. At each time step, the continuation value is estimated using GPR, and a decision tree determines whether to exercise or continue. The method is designed to reduce variance and improve interpretability, particularly in low-dimensional settings. We report the results from (Goudenege et al., 2020, Tables 1–3) using $P = 1000$ training points, which offers the highest reported accuracy despite increased computational cost compared to $P = 250$ or $P = 500$.

**GPR-EI.** GPR with Expected Improvement (EI) follows a sequential design strategy inspired by Bayesian optimization. It actively selects the most informative sample points by maximizing expected improvement in the value function, enabling a more data-efficient approximation of the continuation value. As with GPR-Tree, we report the results with $P = 1000$ training points.

**GPR-MC.** This variant uses GPR to estimate the continuation value within a standard Monte Carlo regression framework. It replaces linear regression with nonparametric GPR to improve accuracy, especially in high-dimensional problems.

**Ekvall.** This baseline method is based on the lattice-based regression approach proposed in Ekvall (1996), which approximates the value function using basis functions and optimal stopping. It serves as a classical benchmark for evaluating newer machine learning-based methods.

**Benchmark.** A closed-form analytical solution is available only for the Geometric Basket Put option.

**Our method.** We kept a basic implementation, exploiting classic libraries. We report the average performance of our method over 10 repetitions, along with corresponding confidence intervals. The regularization parameter is simply set to $\lambda = 10^{-6}$, and the RBF kernel lengthscale is selected from the grid $\{40, 80\}$. Sample sizes increase with dimensionality; for instance: for $d = 2$, we use $n = 200$, $M = 50$; for $d = 20$, we use $n = 800$, $M = 100$. All experiments were run on Google Colab using an NVIDIA T4 GPU (16 GB) with a single Intel Xeon CPU and approximately 12 GB of RAM. The FALKON algorithm (Meanti et al., 2020) is taken from https://github.com/FalkonML/falkon.

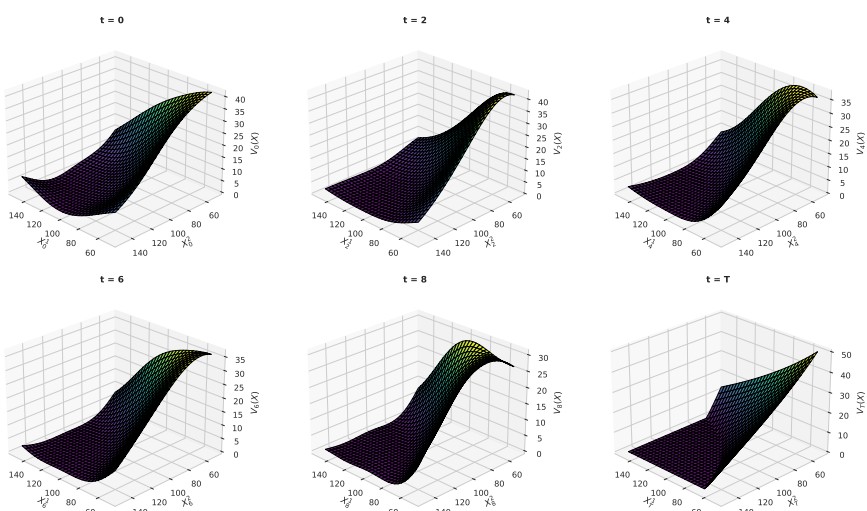

Figure 1: Value function estimates for the Geometric Basket Put option ($d = 2$), see Table 1.

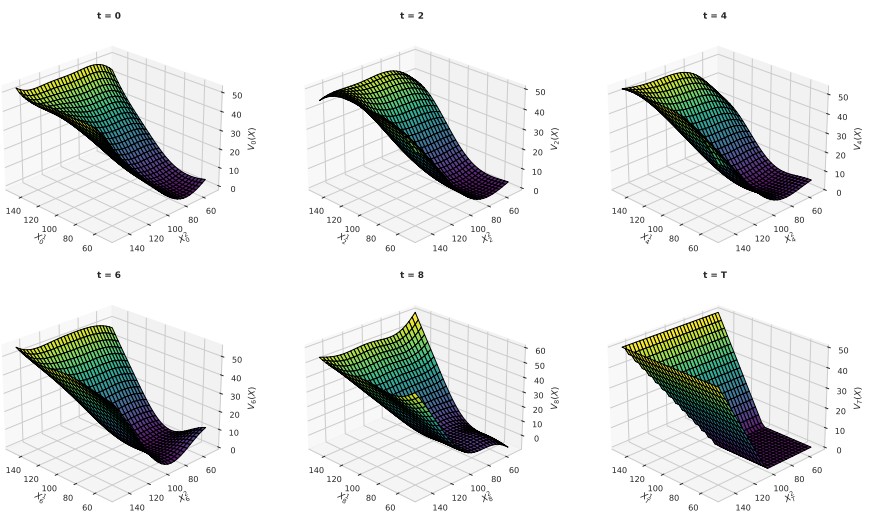

Figure 2: Value function estimates for the Max-Call option ($d = 2$), see Table 2.

# D SUFFICIENT CONDITIONS FOR WELL-POSEDNESS

We discuss here the minimal condition needed for our formulation to be well posed in relation to Assumption 1. Given a function $f \in L^2_{\mu_{t+1}}$, we study under which condition $P_t^u f$ belongs to $L^2_{\mu_t}$, with

$$P_t^u f(x) = \int_{\mathcal{X}_{t+1}} f(x')\, P_t^u(x, dx').$$ (60)

Using Jensen's inequality:

$$\|P_t^u f\|^2_{L^2_{\mu_t}} = \int_{\mathcal{X}_t} \left( \int_{\mathcal{X}_{t+1}} f(x')\, P_t^u(x, dx') \right)^2 \mu_t(dx) \leqslant \int_{\mathcal{X}_{t+1}} f(x')^2 \underbrace{\int_{\mathcal{X}_t} P_t^u(x, dx')\mu_t(dx)}_{=:\, q_t^u(dx')}.$$ (61)

If the pushforward measure $q_t^u$ is absolutely continuous with respect to $\mu_{t+1}$ and admits a bounded Radon–Nikodym derivative, i.e.,

$$\left\| \frac{dq_t^u}{d\mu_{t+1}} \right\|_{L_{\mu_{t+1}}^\infty} \leqslant c_P < \infty, \tag{62}$$

then we obtain:

$$\|P_t^u f\|_{L_{\mu_t}^2} \leqslant c_P^{1/2} \|f\|_{L_{\mu_{t+1}}^2}, \tag{63}$$

which is exactly the requirement in Assumption 1.

Although condition 63 may appear strong, it can often be verified in applications. Indeed, observe that

$$\|f\|_{L_{\mu_{t+1}}^2}^2 = \int_{\mathcal{X}_t} \mathbb{E}\left[f(\pi_t(x, \bar{u}_t(x), Z_{t+1}))^2\right] \mu_t(dx). \tag{64}$$

Therefore, a sufficient structural condition for 63 to hold is the pointwise inequality:

$$\sup_{u \in \mathcal{U}_t} \mathbb{E}\left[f(\pi_t(x, u, Z_{t+1}))\right]^2 \leqslant c_{P,t}^2 \, \mathbb{E}\left[f(\pi_t(x, \bar{u}_t(x), Z_{t+1}))^2\right], \quad \text{for } \mu_t\text{-a.e. } x \in \mathcal{X}_t. \tag{65}$$

This provides a more verifiable condition for establishing Assumption 1, especially in simulation-based settings where the behavior distribution is known or controlled.

