# OpenReview forum: "Error Propagation in Dynamic Programming: From Stochastic Control to Option Pricing"
_ICLR.cc/2026/Conference — Submitted to ICLR 2026_

### Official Review · Reviewer_1Tfg · 2025-10-27

**Soundness:** 4
**Presentation:** 4
**Contribution:** 3
**Rating:** 8
**Confidence:** 4

**Summary:**

This paper develops a framework to analyze error propagation in discrete-time stochastic optimal control solved via a specific form of approximate dynamic programming. For this, the authors combine Monte Carlo estimation and kernel ridge regression within reproducing kernel Hilbert spaces to approximate value functions backward in time.
A detailed error decomposition, into regression, sampling, and propagation errors, leads to non-asymptotic convergence bounds (Theorem 1). The approach is applied to American option pricing, where the proposed algorithm achieves competitive accuracy.

**Strengths:**

The paper is well written and the problem is clearly presented. The structure is easy to follow, starting from the theoretical setup, over the algorithmic formulation, to the numerical illustration. A particular strength is the detailed and rigorous error decomposition, which separates regression, Monte Carlo, and propagation errors, and analyzes each of them carefully. The corresponding theoretical guarantees are well motivated and technically solid. The application to American option pricing is also well described and interesting, showing how the proposed framework can be applied to a practically relevant problem and how the theoretical insights translate into a concrete numerical algorithm.

**Weaknesses:**

While the numerical experiment demonstrates that the runtime of the proposed method is superior to the benchmark algorithms, it would have been valuable to numerically verify some of the theoretical claims, such as the predicted convergence rates or the scaling of the individual error components. This would help assess how tight or meaningful the theoretical bounds are in practice. Moreover, it remains unclear how informative the bound in Theorem 1 actually is for problems with large horizons, since the recursive error term grows with the time horizon $T$. This raises questions about the practical usefulness of the theoretical result for long- or infinite-horizon control problems.

**Questions:**

Questions:

1) It is unclear to me what the assumptions on the state and action spaces are. Reading Section 2 (lines 108–161), it seems they could be very general (e.g., Polish spaces). However, later in the paper compactness of the state space appears to be required, and in bound (26, line 386) the argument seems to rely on a finite action space. This creates confusion. Could you please clearly state the precise assumptions on the state and action spaces already in Section 2 when introducing the problem?

2) I do not agree with the statement :
“Despite its practical relevance, discrete-time SOC has historically received less theoretical attention and often presents greater challenges due to the absence of many of the mathematical tools available in continuous time.”
Please consider the excellent textbooks Discrete-Time Markov Control Processes by Onésimo Hernández-Lerma and Jean-Bernard Lasserre. In my opinion, discrete-time SOCs are extremely well studied, both theoretically and practically.

3) How can one verify Assumption 3 in practice? In particular, how can one determine or estimate the smoothness parameter $\beta_t$?

4) It seems that the Error-Backpropagation term in Theorem 1 grows with the horizon $T$, which raises the question of how useful the result remains for large-horizo (or even infinite-horizon) problems. Could the authors please comment on this limitation?

5) Typos:

a) Eq. (8): should read $\mathcal{U}_t$ instead of $U_t$.

b) Line 260: “$\mathcal{U}$” should likely be $\mathcal{U}_t$.

c) Line 239: “defiend” → “defined.”

---

> ### Author Response · Authors · 2025-11-23
>
> ***Response***
> We thank the reviewer for the comments and address each point below.
>
> ---
>
> 1) The formalization of the problem in Section 2 is intentionally general. Our goal is to set up a mathematical structure with minimal assumptions—such as Assumption 1—to guarantee well-definiteness and stability of the recursion. Additional conditions are then introduced in Section 4, where the objective is to obtain optimal excess-risk bounds and to control error propagation. For example, we introduce the source condition (and, in the appendix, the capacity condition) to bound the regression error in Term I. Likewise, we highlight two classical settings in which a finite upper bound on the empirical Rademacher complexity in Term II is available. These two cases are directly relevant to the financial application considered in the paper (American option pricing corresponds exactly to the finite set of controls case). The complexity control of function classes is a well-established topic in statistical learning theory and functional analysis, and providing a comprehensive treatment of it lies beyond the scope of this work.
>
> ---
>
> 2) Thank you for the comment. We agree that the original sentence was too strong. We have revised the statement accordingly and added the suggested citation alongside the other classical references.
>
> ---
>
> 3) Assumption 3 is a standard source (smoothness) condition in statistical learning theory, and it cannot, in general, be directly verified from data. The smoothness parameter reflects how well the target function aligns with the RKHS associated with the kernel, and this quantity is not observable without additional structural knowledge of the underlying model. In practice, one typically uses data-driven procedures (e.g., cross-validation) to choose regularization parameters without explicitly estimating the smoothness. These methods implicitly adapt to the unknown regularity of the target, even if the underlying source exponent is not identifiable.
>
> ---
>
> 4) It is expected that the total error grows with the horizon $T$, since each additional time step introduces a regression and sampling error that must be propagated backward (similar to regret accumulation in online learning). The purpose of Theorem 1 is precisely to quantify how this accumulation occurs and to show that, with the prescribed choices of $\lambda_t$ and $M_t$, the growth can be controlled in terms of the sample sizes $n_t$.
> Importantly, the factor governing error propagation is $c_P$. In many applications—including the main one studied in the paper (American option pricing)—we have $c_P < 1$, which means that propagated errors are damped rather than amplified as we move backward in time. In such settings, the bound remains stable even for moderately large horizons.
>
> ---
>
> 5) We corrected the mentioned typos.

---

> > ### Comment · Reviewer_1Tfg · 2025-11-26
> > **Response to rebuttal**
> >
> > I thank the authors for their response and again point out that I support acceptance of this paper.

---

### Official Review · Reviewer_a8rA · 2025-10-30

**Soundness:** 2
**Presentation:** 1
**Contribution:** 3
**Rating:** 2
**Confidence:** 4

**Summary:**

This paper considers a regression-based approach to solving discrete-time stochastic optimal control problems. Given a controlled, possibly nonstationary, Markov process, the goal is to design a control law that maximizes the cumulative sum of rewards over a finite horizon $t\in [0, T-1]$ plus a terminal reward at $t=T$. The paper proposes a regression-based approach to solve the dynamic programming problem and analyses the error propagation.

The algorithm is designed by first formulating the Bellman equation, $V_t(x)=ess sup_u F_t(x,u) + E[V_{t+1}(X_{t+1}^u)\mid X_{t}^u=x]$. To solve for the value function at time $t=0$, the algorithm starts from $t=T$ (where $V_T$ is known), and at each time step estimates $V_{t}$ using kernel ridge regression (KRR) with simulated data. The data is obtained by sampling input states according to the dynamics $x_t\sim \mu_t$ and generating the corresponding labels $y$ using the Bellman equation, where the expectation term is approximated with Monte Carlo (MC) simulation and $V_{t+1}$ is replaced with the function estimate $\hat{V}_{t+1}$ from the preceding time step.

To analyse the error $\|\hat{V}_0-V_0\|$, the authors start by providing an upper bound of the error $\|\hat{V}_t-V_t\|$ consisting of the sum of three naturally arising errors: the KRR error, the MC approximation error, and the propagation error associated with the Bellman operator. The KRR error scales as a function of the number of data samples $n_t$ and a parameter $\beta_t$ which captures how well the value function can be approximated by an estimator from the RKHS. The MC error can be made to scale as the KRR error for a sufficiently large number of MC samples. The propagation error amplifies the error at the preceding time step with the Lipschitz constant $c_P$ of the Bellman operator. The final upper bound $\|\hat{V}_0-V_0\|$ therefore scales as a function of $n_t$, $\beta_t$, and $c_P$.

To verify the theory, the authors prove two auxiliary lemmas in addition to leveraging well-known results in statistical learning theory. The authors provide a running example with the pricing of an American max-call option, and evaluates the optimal policy under the algorithm in such a setting while comparing with a theoretical benchmark and baseline methods.

**Strengths:**

The problem setting is rigorously formulated and the background on stochastic optimal control is well-explained. The proposed algorithm is relatively simple to understand, and the chosen means of function approximation (KRR) is motivated from the perspective of being able to derive error bounds. The error decomposition is natural (although somewhat arbitrary, since it is formed by adding and subtracting relevant terms and using the triangle inequality). To the best of my knowledge, the theoretical results are correct in most cases (see “Weaknesses” for exceptions), even if it is unclear how much they inflate the upper bound on the error. I appreciate the running example on American-style max-call options, which clarifies the motivation (that traditional methods break down for sufficiently large baskets) and offers a nice illustration the methodology resulting in simple expressions.

The paper is generally clearly written and follows a logical structure.

The proof of Lemma 1 is correct from my understanding. While Lemma 2 holds, the proof has a significant mistake, I believe (see “Weaknesses”).

The experiments show that the algorithm yields

**Weaknesses:**

I believe there are major weaknesses: 1) significant error in a proof, 2) incorrect citation format, 3) missing experimental evidence of how the algorithm can be designed to control the error by inducing favorable values of key parameters ($n_t, \beta_t, c_P$), and 4) missing theoretical discussion on appropriate kernel design (which defines the RKHS) given the form of the value function.

It is difficult for the reader to understand the usefulness of the error bounds. The authors claim that the error bounds can be used to control the error (L057); however, this is not experimentally verified. The paper would benefit from an experimental evaluation of how the error relates to the key parameters identified in the error analysis, such as $c_P$ and $\beta_t$, by repeating experiments under settings reflecting varying $c_P$ and $\beta_t$. For example, $\beta_t$ might be controlled by choosing more or less appropriate kernels given the reward function.

I am missing a more detailed discussion on the choice of kernel, and thus the range of $\beta_t$, given the form of the value function, generally or for specific reward functions. For example, in the American call option example, the authors note that $V_T$ is non-smooth and $\beta_t$ thus can be small (which inflates the error bound), yet proceeds to use the RBF kernel in experiments which represents smooth functions and therefore is unsuitable.

The proof of Lemma 2 has a mistake, although the conclusion still holds. Specifically, in proving the Lipschitz property, the authors claim that $\|ess sup_u [F_t(\cdot, u)+P_t^u g] – ess sup_u [F_t(\cdot, u)+P_t^u f]\|=\|ess sup_u [ P_t^u g –P_t^u f]\|$, but if suffices to consider a binary control with a pretty arbitrary lookup table for $P_t^ug, P^u_tf$ in the special case that $X^u_{t+1}$ is independent of $X^u_t$ to see that it does not hold. I think it should be an inequality sign, in which case the final inequality still holds.

In terms of presentation, the paper is not using the correct ICLR citation convention which separates between \citet, \citep. Instead, the authors exclusively use \citet, while \citep is the correct option in many places. Statements such as L039, *“While continuous-time SOC has been extensively studied in the literature Bertsekas (2012), […], where decisions are made at fixed time intervals Bertsekas & Shreve (1996); Puterman (2014).”*, has a significant effect on readability. Especially the paragraph L058-L072 is tough to read as result.

*Minor comments (shared for the benefit of the authors, not affecting my assessment)*

In Theorem 1, the authors refer to the decay rate $\lambda_t\sim n_t^{-1/(\beta_t+1)}$. This arises from a corollary in Appendix B, and the main text is missing a reference to this result or an explanation.

Typo/grammatical error in L054, L065, L239.

The equation notation is inconsistent, see L158 vs L202.

**Questions:**

In Algorithm 1, how would you generate the input values $x_t\sim \mu_t$ when the control influences the dynamics of $x_t$  beyond sending it to the cemetery state? In the case of the American option, the dynamics of $x_t$ is effectively independent of $u$, where $u$ just sends it to the cemetery state when activated. So here I see that forward simulation of the SDE (Eq. 19) would work. It is not clear to me how to implement the algorithm, which starts from $T$ and goes backward, when the SDE contains a general function of $u$ (which $\pi_t$ allows).

What fundamentally separates your method from benchmark methods based on Gaussian Process Regression (GPR), e.g., GPR-MC, considering that GPR can be viewed as KRR (such that the posterior mean function relates to the KRR function estimate)?

---

> ### Author Response · Authors · 2025-11-23
>
> ***Response***
> We thank the reviewer for the comments and address each point below.
>
> ---
> - The derived bounds play the role of classic excess risk bounds in statistical learning theory, extended here to account for time propagation.
> Equations (22)–(24) provide the standard generalization error of the estimator under the square loss, and Eq. (28) shows precisely how these errors propagate to determine the final accuracy on V_0.
> We agree that systematic experiments varying c_P, β_t, or kernel families would further illustrate the robustness of the analysis. However, such a study would require controlled synthetic SOC environments where these parameters can be tuned—something that goes beyond the scope of this theoretical paper. Our limited simulations are intended only to show that the proposed method performs reliably on the main application we had in mind, namely standard American option pricing (for which Table 1 also includes a theoretical benchmark), illustrating the practical soundness of the framework.
> Given the limited space and the theoretical focus of this work, a more extensive empirical investigation—including controlled variation of c_P(e.g., through discounting), β_t(e.g., smoother or less smooth regression targets), a competitive implementation of our algorithm, and a broader comparison with other methods and applications—is left for follow-up work.
>
> ---
>
> - We intentionally did not discuss specific kernel choices, since our focus is *theoretical*.
>   Our goal was to establish general statistical guarantees valid for any bounded kernel, rather than to optimize kernel selection for a particular task.
>   In the experiments, we deliberately used the *RBF kernel*, the most common choice in practice, even though in this specific application—where the payoff $V_T$ is non-smooth—it represents the *misspecified case* ($\beta_t<1$).
>   This allows us to empirically illustrate the theoretical framework discussed in Section 4, beyond the classic favorable well-specified regime ($\beta_t=1$).
>   A specific choice of kernel and its impact on $\beta_t$ in different applications is left for future work (see also answer to reviewer z7ZE).
>
> ---
>
> - Thanks for noticing the inequality LaTeX typo in the proof.
>   As correctly pointed out, and as stated in line 755, we simply apply the inequality introduced just above (Eq. (33) in Lemma 1), which is indeed an inequality, not an equality.
>   No further changes are needed since the typo was limited to that line, and Lemmas 1 and 2 were already correctly stated.
>
> ---
>
> - Thanks for pointing out the ICLR convention on citations.  We have updated all references to use the correct "\citet" and "\citep" formats.
>
> ---
>
> - The choice of $\lambda$ is discussed in L856.
>   We added a precise pointer to the corresponding Corollary in Appendix B before Theorem 1 and corrected minor grammar and referencing issues.
>
> ---
>
> - **Sampling when dynamics depend on control**
>   We thank the reviewer for this insightful question.
>   This issue is common in reinforcement learning and dynamic programming, where dynamics depend on the control.
>   As noted in L145–148, a standard solution is to use a *behavior policy* (Sutton & Barto, 1998).
>   If the dynamics $x_{t+1}=\pi_t(x_t,u_t,z_{t+1})$ depend on $u_t$, then $\mu_t$ would depend on the policy being learned, creating circularity if defined under the optimal policy.
>   Instead, one fixes a non-optimal, possibly random behavior policy $\bar u_t$ and simulates trajectories forward:
>
>   $$X_{t+1} = \pi_t(X_t, \bar u_t(X_t), Z_{t+1})$$
>
>   defining $\mu_t$ as the empirical state distribution at each time.
>   During the backward phase, these samples $x_t\sim\mu_t$ are fixed, and for each $x_t$ we compute targets by simulating next states under all actions $u$:
>
>   $$y_i = \max_{u\in\mathcal U_t}\Big(F_t(x_i,u) + \frac{1}{M}\sum_j \widehat W_{t+1}(\pi_t(x_i,u,z_j))\Big)$$
>
>   This allows exploring any control $u$ through $\pi_t$ without altering $\mu_t$.
>   As is standard in RL, we assume that the support of the behavior policy covers that of the target policy, which corresponds to the absolute continuity condition discussed in Appendix D and linked to Assumption 1.
>
> ---
>
> - **Comparison with other methods**
>   We agree that GPR can be used to obtain the KRR estimator.  However, our contribution is not a new regressor, nor simply the classic idea of iterating approximations backward in time.  As stated at the end of Section 1, our goal was to build a *general and well-structured RKHS framework for discrete-time stochastic optimal control*, featuring a backward DP scheme with an explicit *error decomposition* (regression, Monte Carlo, Bellman propagation) and *finite-sample rates* under general source and capacity conditions. This is what mainly differentiates our work from Goudenege et al. (2020) and other numerical papers.

---

> > ### Comment · Reviewer_a8rA · 2025-11-27
> >
> > Thank you for updating the proof, revising the citation format, and adding a reference to the Corollary in Appendix B which is relevant for L856.
> >
> > Thank you also for clarifying how to deal with the circularity problem.
> >
> > I appreciate the rigorous theoretical analysis of the paper, but given that the error bound is a central part of the contribution, I still think that the paper would benefit from an experiment that illustrates its properties – just a simple, fully synthetic (toy) experiment would suffice. In my view, it would clarify the impact especially of the misspecification parameter $\beta_t$, which would help anyone who wants to apply your method to their problem.
> >
> > Regarding comparison with other methods, I understand that your proposition is the RKHS framework approach to DT-SOC with error decomposition. I think it would be useful to have a more detailed description of how your approach is different from the GPR-based benchmarks. When I read the descriptions in Appendix C, it is not clear to me why your approach has a conceptual advantage  over the baselines (beyond the error decomposition). Perhaps you can illustrate this in a simple figure? This is not a major limitation though and it does not affect my assessment, it is just a suggestion.
> >
> > I am happy to update my ranking as follows (was not able to edit my previous grades above):
> >
> > Soundness
> > 3 (2)
> >
> > Presentation
> > 3 (1)
> >
> > Contribution
> > 3
> >
> > Rating
> > 4 (2)

---

> > > ### Author Response · Authors · 2025-12-03
> > >
> > > We thank the reviewer for the reply.
> > > We agree that illustrating the effect of the misspecification parameter $\beta_t$ can be informative. However, constructing a synthetic SOC environment where $\beta_t$, $c_P$, and the source/capacity conditions can be independently controlled through time is technically nontrivial. These quantities arise from the interaction between the transition dynamics, the reward structure, and the Bellman operator, and cannot be easily tuned. This difficulty is well known in statistical learning theory: source and capacity conditions are rarely objects one can “set” directly, even in standard regression, let alone within a recursive DP structure.
> > > We will add some additional description in Appendix C of the comparison methods. We refer to the above answer regarding the difference between these methods and our approach. Thanks for the suggestions.

---

### Official Review · Reviewer_mxZL · 2025-10-31

**Soundness:** 2
**Presentation:** 3
**Contribution:** 2
**Rating:** 4
**Confidence:** 3

**Summary:**

This paper investigates the theoretical and methodological foundations for solving discrete-time stochastic optimal control problems. The authors formulate the control problem within a general dynamic programming framework and propose an approximation algorithm that combines nonparametric regression with Monte Carlo  sampling. Specifically, it employs Kernel Ridge Regression within a RKHS for value function estimation and MC sampling to estimate the continuation value.

**Strengths:**

1. **explicit error propagation bound**: The paper’s central theoretical contribution lies in rigorously quantifying how regression, sampling, and propagation errors interact and accumulate backward in time. This explicit and interpretable result provides a understanding of error propagation in DP-based SOC learning.

2. **good presentation**: Overall, I really enjoyed reading this paper. The authors present the main problem clearly and structure the exposition in a logical and accessible manner.

3. **Time relevant reserach topic**: Rigorous analysis for SOC is a challenging research area. The paper's value lies in its attempt to directly address this critical problem, aiming to provide a complete end-to-end theoretical guarantee for a specific (kernel-based) methodology.

**Weaknesses:**

1. **Lack of justification for KRR and ambiguous contribution positioning**: The choice of KRR as the main regression tool is not convincingly justified. The paper implicitly positions KRR as theoretically superior by emphasizing its analytical tractability, while underestimating the rich theoretical results already available for neural networks in similar settings. For instance, neural network–based approaches have achieved strong theoretical guarantees in high-dimensional SOC, PDE approximation, and deep reinforcement learning.

[1] Tsang, Ka Ho, and Hoi Ying Wong. Deep-learning solution to portfolio selection with serially dependent returns. SIAM Journal on Financial Mathematics 11.2 (2020): 593–619.

[2] Lim, Dong-Young, et al. Langevin dynamics based algorithm e-THEO POULA for stochastic optimization problems with discontinuous stochastic gradient. Mathematics of Operations Research 50.3 (2025): 2333–2374.

[3] Huré, Côme, et al. "Deep neural networks algorithms for stochastic control problems on finite horizon: convergence analysis." SIAM Journal on Numerical Analysis 59.1 (2021): 525-557.

[4] Beck, Christian, et al. "Deep splitting method for parabolic PDEs." SIAM Journal on Scientific Computing 43.5 (2021): A3135-A3154.

[5] Neufeld, Ariel, Philipp Schmocker, and Sizhou Wu. "Full error analysis of the random deep splitting method for nonlinear parabolic PDEs and PIDEs." arXiv preprint arXiv:2405.05192 (2024).

2. **Strong assumption (Asumption 1)**: The second condition in Assumption 1 appears quite restrictive, as it conveniently simplifies the propagation error.

3. **Moderate theoretical novelty**: Consequently, Theorem 1 essentially aggregates three well-known components: (1) a standard KRR estimation bound, (2) a standard Monte Carlo sampling bound, and (3) a propagation bound derived almost trivially from Assumption 1. As such, the overall theoretical novelty may be limited.


4. **Scalability issue**: Kernel methods are known to suffer from the curse of dimensionality, making their application to high-dimensional SOC settings challenging. The experiments are restricted to small-scale problems, and no comparison with deep learning–based methods is provided.

**Questions:**

1.  If the error propagation (Term III) is fundamentally determined by the MDP's properties ($c_P$) and not the regression model, what is the fundamental theoretical advantage of using KRR, and accepting its critical scalability issues, over neural networks, whose approximation and generalization bounds are also well-studied? Does the explicit convergence rate for Term I obtained via KRR justify the cost of curse of dimensionality?

2. Beyond discounted option pricing, under what classes of SOC models does $c_p\leq 1$ hold?

3.  How does the use of the FALKON approximate solver in Section 5 align with the theoretical analysis in Section 4?

4. It would be benefittial to include comparisons with deep learning–based approaches under the same DP setting.

5. Can this approach be extended to high-dimensional state spaces (e.g., 100 dimensions)?

---

> ### Author Response · Authors · 2025-11-23
>
> ***Response***
> We thank the reviewer for the comments and address each point below.
>
> ---
>
> 1) We acknowledge the rapid progress in the theoretical analysis of neural-network–based methods for SOC, PDEs, and reinforcement learning. Our choice of KRR is not based on claiming kernel methods are universally superior, but on the goals of this work: obtaining explicit, non-asymptotic, and easily interpretable error bounds in a general discrete-time SOC setting.
> KRR provides closed-form estimators, a transparent bias–variance decomposition, and sharp minimax-optimal excess-risk rates (Caponnetto & De Vito, 2007).
> Our aim is not to compare kernels with neural networks but to build a clean theoretical framework for discrete SOC problems. We view our approach as a basis upon which analogous results for other models can be studied (see lines 339-348).
>
>
> ---
>
> 2) We agree that the second condition in Assumption 1 is arguably strong, but it is not introduced merely to “conveniently” simplify the propagation error. This assumption is considerably weaker than the common assumption in the literature of uniformly bounded utility functions and compact state spaces, which would imply our $L^2$ boundedness assumption.
> This assumption is essential for two reasons. First, it guarantees that all objects in the Bellman recursion remain in $L^2_{\mu_t}$, ensuring that the dynamic programming operators are well defined at every step. Second, it provides the stability needed to control how errors propagate backward in time, preventing uncontrolled amplification in the recursion.
> Importantly, as shown in the paper, the assumption holds in the main application motivating this work (American option pricing). Since this is not always guaranteed, we further discuss the assumption in Appendix D, including a more practical way to verify it in concrete problems.
>
> ---
>
> 3) Indeed, the novelty of our work does not lie in the individual methods involved (such as KRR, which is extensively studied), but rather in the mathematical RKHS-based formulation of the SOC problem, the proposed error decomposition and its individual contributions, and the resulting excess-risk convergence with a precise application to the important problem of American option pricing.
>
> ---
>
> 4) We used part of the additional space to extend the experiments to dimensions 20 and 40. As expected, increasing the dimension noticeably slows down our simple implementation. This slowdown is not mainly due to the regression term but especially due to the Monte Carlo component. We commented on this at the end of Section 5. Possible improvements were also mentioned in the conclusions, where we identify the Monte Carlo step as the principal bottleneck for scalability. A natural direction for a competitive follow-up implementation is to exploit the modularity of our approach to replace Monte Carlo sampling with more efficient methods such as quadrature or monomial rules.
> As for the first term (the regression term), kernel methods have already been shown to be competitive against neural networks on certain types of data and problems (see, e.g., the FALKON paper, Meanti et al. 2020). Moreover, comparing different regression methods is not the goal of our work—one may simply substitute their preferred regressor and use its statistical guarantees in Term I.
>
> ---
> Questions:
> ---
>
> 1) See point 1 above. Further, positioning kernel methods against neural networks is not the scope of this work. Regarding scalability, modern projection-based kernel approximations (Nyström, random features, FALKON, etc.) have been shown to be empirically competitive—and in some regimes even superior to neural networks depending on the data structure (e.g., Meanti et al. 2020). Thus, using kernels does not inherently preclude scalability.
>
> ---
>
> 2) Beyond option pricing, the condition $c_P < 1$ holds in a wide class of discrete-time SOC models, including discounted MDPs with discount factor $\gamma < 1$,
> \[
> T f(x) = \max_{u} \left\{ r(x,u) + \gamma \mathbb{E}[f(X') \mid x,u] \right\},
> \]
> economic models with discounted utility or revenue, or dissipative physical systems (e.g., energy storage).
>
> ---
>
> 3) See answer to reviewer z7ZE and new lines 339–344 and 490–498.
>
> ---
>
> 4) We agree that a competitive implementation of our algorithm, together with a broad comparison with other methods and applications, is a natural follow-up of this work. We remark that, as clearly stated in the paper, our contribution is theoretical. Moreover, substituting a deep-learning method for KRR in Term I would simply reduce to comparing neural networks with kernel methods on a regression task—a well-studied research direction and not the scope of this paper.
>
> ---
>
> 5) This is discussed at point 4) above.

---

> > ### Comment · Reviewer_mxZL · 2025-11-27
> >
> > Thank you for the rebuttal.
> >
> > The rebuttal clarifies intent but does not resolve the fundamental concerns regarding scalability, novelty, and generality. While the error decomposition is clean and useful, the contribution remains incremental and limited in practical scope.
> >
> > In particular, if the paper aims to make a primarily theoretical contribution rather than addressing scalability or practical relevance, it should provide a new proof technique or a clear theoretical advancement. However, the analysis relies on standard minimax bounds and a restrictive assumption for stability propagation, resulting in a main theorem that does not yield a fundamentally new analytical insight beyond existing frameworks.
> >
> > Therefore, I will keep my original evaluation and recommendation.

---

> > > ### Author Response · Authors · 2025-12-02
> > >
> > > We thank the reviewer for the answer. We would like to clarify again the nature and novelty of our contribution.
> > >
> > > Our main contribution is **not** a new equivalent bound for classic regression or a new proof technique for kernel methods (alternative to already existing optimal ones in Caponnetto, De Vito (2007) or Steinwart et al. (2009) etc) —nor was this the objective of the paper. The novelty lies in the **mathematical formulation of discrete-time SOC within a general RKHS framework**, together with a **fully modular and minimal-assumption error decomposition** for approximate dynamic programming via backward induction, to finally study the propagation of the error backward in time. To the best of our knowledge, such a formulation and decomposition do not appear in the existing literature.
> > >
> > > We also emphasize that we work under **minimal and substantially weaker assumptions** than those commonly used in SOC/MDP analyses. As mentioned above, Assumption 1 is much lighter than the standard assumptions of uniformly bounded utility functions and compact state spaces (which would immediately imply our $L^2$ boundedness condition). In addition, our results explicitly handle the **misspecified regime**, whereas much of the classical analysis focuses on the well-specified case.
> > >
> > >
> > > Within this general framework, we incorporate **adaptive statistical-learning assumptions** (source and capacity conditions), which allow the theory to adjust to the regularity of the problem. The resulting error decomposition—cleanly separating regression, sampling, and propagation terms—enables the use of *any* function approximator with known generalization guarantees. The reviewer notes that the regression bound is standard; this is correct, and precisely the point: the novelty is in **how these classical components are assembled, propagated, and controlled** in a rigorous DP context. The contribution is therefore conceptual and structural, not a re-derivation of minimax bounds for regression.
> > >
> > > Given these minimal and widely used assumptions, we are not sure why the contribution is described as “limited in practical scope.” This probably refers to the previously cited scalability issues of exact kernel methods compared to neural networks. While comparing them is not the focus of this paper, we note that modern randomized kernel approximations (Nyström, random features, FALKON) achieve near-linear complexity and have been shown to be competitive with—and in some settings superior to—neural networks, including in financial applications. Our framework naturally accommodates these scalable variants (see lines 339–348 and 490–497), since Term I can be replaced by any regression method with equivalent statistical guarantees.
> > >
> > > The practical experiments included are intentionally limited by the 9-page constraint and by the theoretical goals of the paper. We used the additional space to enlarge our simulations to higher dimensions as a preliminary analysis of the scalability properties of the method. A systematic study of scalability, engineering optimizations, comparisons with deep RL/control methods, and applications beyond option pricing is planned for a dedicated follow-up work with an explicitly numerical and experimental focus. The present submission aims to establish the **foundations**—a rigorous mathematical structure, minimal assumptions, and explicit propagation bounds—on which such applied developments can be built.

---

### Official Review · Reviewer_z7ZE · 2025-11-01

**Soundness:** 3
**Presentation:** 3
**Contribution:** 3
**Rating:** 4
**Confidence:** 4

**Summary:**

Paper describes how approximation errors (from function approximation and sampling) propagate backward  through discrete‑time stochastic control / dynamic programming (DP), and do so in a mathematically clean way that scales to practical Monte Carlo pipelines.
Cast the value‑function recursion in (L^2) spaces and approximate each stage with Kernel Ridge Regression (KRR) in an RKHS, while estimating conditional expectations by Monte Carlo. The paper introduces a **three‑term error decomposition** (regression, sampling, propagation) and derives rates under **model misspecification** (source conditions). It then specializes the analysis to American option pricing and gives a simple, fast implementation using FALKON

**Strengths:**

The paper provides a transparent decomposition of where error comes from and how it composes across time. This makes it easier to reason about sample allocation per stage and to justify backward pipelines widely used in practice.
Many DP papers state rates in well‑specified cases; here the source‑condition analysis (and the no smoothing under max) keeps expectations realistic for options and other kinked payoffs.

**Weaknesses:**

The square‑integrability + bounded pushforward density  (Assumption 1 and App. D) are natural for options (and are verified), but for general controlled Markov dynamics—especially continuous actions—the paper relies on either a finite control set or Lipschitz transitions on compact sets to get the (\widetilde O(M^{-1/2})) MC rate . The continuous‑action coverage is thus limited to fairly regular models and compact truncation.

The clipping/truncation device (Eq. 18 and discussion just after) is standard but introduces bias,  whose impact is not quantified end‑to‑end, in the main bound.

The best‑case rates depend on spectral decay, but the main theorem states the weakest capacity version. While fine for readability, practitioners may want *guidance on kernels / data distributions that realize the faster rates.

The experiments rely on FALKON (Nyström‑sketched KRR), but the theory is for exact Kernel Ridge Regression (KRR.) There is no explicit error budget split showing how sketching error composes with the three terms in Eq. (23).

Comparisons focus on GPR‑based methods; classical LSMC (Longstaff–Schwartz) and primal/dual methods are acknowledged in the related work but not benchmarked. Since those are the workhorses in practice, including them (even on (d\leq 10)) would make the case more compelling. (The paper itself lists these families in Sec. 1.)
   * **Hyperparameter protocol** is light. Tables 1–2 report strong timings, but the **number of training points P** used by GPR‑Tree/EI (1 000) is fixed from a prior paper, while KRR‑DP uses **much smaller (n,M)**; it would help to **sweep (n,M)** to show price‑error vs time trade‑offs and confirm the speedup is not just from fewer samples (App. C gives some sizes).
   * All experiments use **lognormal GBM** with (T=9), fixed (\sigma,\rho). Stress‑tests (larger (T), heterogeneous vols, path‑dependent payoffs) would probe the **error‑propagation** claims more thoroughly.

5. **No adaptive allocation policy**
   The theory suggests how errors enter, but the algorithm does not exploit it to **adapt ((n_t,M_t,\lambda_t))** to **measured difficulty** (e.g., smaller (\beta_t) near maturity). An **adaptive budget** could yield tangible gains.

6. **Missing constants / finite‑sample guidance**
   The **“with high probability”** bounds do not expose constants, so direct **sample‑size planning** per stage remains heuristic. A **worked numerical sizing** using the elements in App. B would be valuable.

**Questions:**

Tighten the algorithm–theory link by incorporating a sketched‑KRR error term into Eq. (23) and recommending landmark sizes that keep it below the regression term.
Add baselines beyond GPR: LSMC, a primal‑)dual method, and a basic deep BSDE / deep stopping approach [5,10]. Even a small table would broaden the empirical takeaways.
Report ablations: curves of **price error vs. (n_t), (M_t), (\lambda_t)** and a head‑to‑head **time–accuracy frontier** with GPR‑MC. (The text notes that sample sizes grow with (d) in App. C; plotting their impact would be instructive.)
Quantify clipping/truncation bias in a lemma (e.g., tails under GBM), or show empirically that relaxing bounds does not change prices beyond CIs.
Discuss continuous‑action SOC beyond stopping: explicitly state how the Lipschitz/compact assumptions ensure Eq. (26), and add one small continuous‑control example to illustrate.

---

> ### Author Response · Authors · 2025-11-23
>
> ***Response***
> We thank the reviewer for the comments and address each point below.
>
> ---
>
> Regarding Assumption 1, we refer to our response to reviewer mxZL, point 2. Concerning the reviewer’s remark about assuming a finite control set or Lipschitz transitions on compact sets, note that such conditions relate instead to the Monte Carlo Term II. In particular, they are required to control the empirical Rademacher complexity of the function class induced by the transition dynamics. Complexity control for function classes is a classical topic in statistical learning theory and functional analysis, and it lies outside the scope of this work.
>
> ---
>
> The effect of clipping is indeed accounted for in our bounds. The additional clipping operation, given its properties (see line 257), does not degrade the optimal rate of the underlying unclipped estimator, as shown in standard results (e.g., Steinwart & Christmann, 2008). In fact, the excess-risk bound appearing in Term I already corresponds to the clipped estimator.
>
> ---
>
> Fast-decay rates under additional capacity assumptions are discussed in Appendix B. Given space limitations, we chose to present the most general capacity-free result in the main text and defer these refinements to the appendix. The faster rates require a standard spectral-decay assumption on the kernel–distribution pair, which is not determined by the kernel alone but depends jointly on the kernel, the domain, and the sampling distribution. Moreover, these refinements depend on the interplay between capacity and source conditions—a classical topic in statistical learning theory that lies beyond the scope of this work.
>
> ---
>
> We do not discuss Nyström or other random projection methods (such as random features) in detail, since our main focus is on how the error propagates through time after solving a sequence of KRR problems. In the experiments, we employ FALKON as a fast solver, but note that this type of approximation can be incorporated seamlessly into our framework. The Nyström estimator is known to preserve the statistical guarantees of exact KRR (see Rudi et al., 2017; Della Vecchia et al., 2024), so it can be directly substituted in the bound without altering the convergence rate.
> A key advantage of our approach is the modularity of the error decomposition: each contribution is clearly separated, so replacing KRR with a Nyström approximation affects only Term I (while maintaining the same rate) and leaves the remaining error terms—and the overall propagation analysis—unchanged. We used the additional space to add brief explanations of this in lines 339–348 and 490–497.
>
> ---
>
> As explicitly stated in the paper, our contribution is primarily theoretical—its core focus is statistical learning theory for dynamic programming. The empirical comparisons included are intentionally preliminary and serve only to validate the soundness of the approach on a standard benchmark such as American option pricing. The method itself is general and can be applied to other settings; more extensive empirical evaluations—e.g., larger horizons $T$, heterogeneous volatilities, path-dependent payoffs to assess robustness, as well as applications beyond option pricing to test versatility—are planned for future work.
> Given the limited space and the theoretical emphasis of the paper, our focus here was to establish foundational guarantees and statistical properties of the method in its basic form, preparing the ground for future extensions. We used the available space to extend our numerical tables to higher dimensions (see updated Tables 1 and 2 and lines 499–505). As already anticipated in the previous version of the conclusions, increasing the dimension slows down the algorithm, with the Monte Carlo component becoming the main bottleneck. This points naturally toward future improvements, as discussed in lines 502–505—for example, replacing standard Monte Carlo with more sophisticated methods such as quadrature or monomial rules.
>
> ---
>
> Theorem 1 already provides prescriptions for choosing $\lambda_t$ and $M_t$ to control the respective error terms. As is standard in statistical learning theory, we present clean results in terms of asymptotic rates in $n_t$, up to universal constants: these constants are typically unknown in practice, and the main objective is to understand how the sample sizes $n_t$ and horizon $T$ govern the overall error.
> Regarding adaptive allocation, such strategies would require knowledge of problem-dependent quantities such as $\beta_t$ (the regularity of the regression target at each step) and how it evolves through time (see discussion in lines 415–428)—quantities that, in general, are not known in practice.
>
> ---
>
> Questions: see the above points. See extensions in the tables (d=20, d=40) and the additional comments in orange. See answer to reviewer 1Tfg point 1).

---

### Author Response · Authors · 2025-11-23

***Final Comment to All Reviewers***
We thank all reviewers for their thoughtful feedback and constructive suggestions. We have incorporated clarifications and improvements throughout the paper; all modifications are highlighted in orange for ease of reference.

We would like to kindly reiterate the primary scope of this work. The paper is fundamentally **theoretical**, and the intended subject area is **statistical learning theory**—applied to the analysis of **discrete-time stochastic optimal control** (ICLR Primary Area: learning theory). Our contributions lie in establishing a rigorous functional-analytic framework, introducing a clear and fully modular *error decomposition*, and deriving explicit **excess-risk bounds** together with a precise **propagation-error rate** across time steps. The assumptions we adopt (source conditions, capacity considerations, stability requirements, etc.) are standard in learning theory and are essential for obtaining these guarantees.

Because of this theoretical focus, the simulations included in the paper are deliberately limited: they serve only to validate the *soundness* of the approach on a canonical benchmark (American option pricing). Developing a fast and competitive implementation, performing large-scale engineering optimizations, and conducting broad empirical comparisons across alternative state-of-the-art methods and applications are natural and valuable extensions of this work, but they fall outside the scope of this conference paper and are planned for a dedicated follow-up contribution with an explicitly numerical and experimental focus.

We again thank the reviewers for their insightful comments and for helping us improve the clarity and quality of the paper.

---

### Meta-Review · Area_Chair_fQox · 2025-12-31

**Summary:**

In this paper, the authors describe how to analyze error propagation in discrete-time dynamic programming (DP) to solve stochastic optimal control (SOC). In this analysis, they use Monte Carlo for estimating conditional expectations and approximate each stage of DP with kernel ridge regression in a RKHS. The result is a non-asymptotic error decomposition bound consisting of three‑terms: regression, sampling, and propagation. Finally, they apply their method to American option pricing, where the proposed algorithm achieves competitive accuracy.

The reviewers appreciate the rigorous theoretical analysis of the paper, whose main results are a RKHS-based formulation of the discrete-time SOC problem and an error propagation bound for approximate DP. However, most of them are concerned about the significance of the novelty and contributions of the work. Authors improved the quality of their work by addressing/clarifying several issues raised by the reviewers during the rebuttals, but some of their suggestions, including a thorough comparison (both empirically and theoretically) with similar existing methods, establishing connections between their theoretical results and algorithm, and a discussion on the necessity and strength of the assumptions, remained unanswered. I would recommend the authors to address these issues, expand the scope of their results from a purely theoretical work, and prepare it for an upcoming venue.

**Reviewer Concerns:**

The authors did not properly addressed Reviewer mxZL's concerns regarding the generality and novelty (standard proof techniques and assumptions) of the approach. The reviewer still thinks that the contribution of this work is incremental and of limited practical scope.

The authors partially addressed Reviewer a8rA's concerns. However, concerns related to comparison with other methods (a detailed description) and simple experiments that demonstrate the properties of the error bound remained unaddressed. The reviewer is not convinced about the advantage of the proposed approach over the baselines.

**Reviewer Scores:**

Reviewer z7ZE did not participate in the discussion. Reviewers mxZL and 1Tfg responded to the authors, but decided to keep their initial scores of 4 and 8, respectively. Finally, Reviewer a8rA raised their score from 2 to 4 after reading the authors' response.

---

### Decision · Program_Chairs · 2026-01-26

Reject